# Loss of microRNA-128 promotes cardiomyocyte proliferation and heart regeneration

Wei Huang[1,2], Yuliang Feng[1], Jialiang Liang[2], Hao Yu[2], Cheng Wang[3], Boyu Wang[4], Mingyang Wang[5], Lin Jiang[2], Wei Meng[6], Wenfeng Cai[2], Mario Medvedovic[7], Jenny Chen[7], Christian Paul[2], W. Sean Davidson[2], Sakthivel Sadayappan [8], Peter J. Stambrook[9], Xi-Yong Yu [1] & Yigang Wang[2]

The goal of replenishing the cardiomyocyte (CM) population using regenerative therapies following myocardial infarction (MI) is hampered by the limited regeneration capacity of adult CMs, partially due to their withdrawal from the cell cycle. Here, we show that microRNA-128 (*miR-128*) is upregulated in CMs during the postnatal switch from proliferation to terminal differentiation. In neonatal mice, cardiac-specific overexpression of *miR-128* impairs CM proliferation and cardiac function, while *miR-128* deletion extends proliferation of postnatal CMs by enhancing expression of the chromatin modifier SUZ12, which suppresses *p27* (cyclin-dependent kinase inhibitor) expression and activates the positive cell cycle regulators Cyclin E and CDK2. Furthermore, deletion of *miR-128* promotes cell cycle re-entry of adult CMs, thereby reducing the levels of fibrosis, and attenuating cardiac dysfunction in response to MI. These results suggest that *miR-128* serves as a critical regulator of endogenous CM proliferation, and might be a novel therapeutic target for heart repair.

[1] Key Laboratory of Molecular Target and Clinical Pharmacology, School of Pharmaceutical Sciences & Fifth Affiliated Hospital, Guangzhou Medical University, Guangzhou, Guangdong 511436, China. [2] Department of Pathology and Laboratory Medicine, University of Cincinnati College of Medicine, Cincinnati, OH 45267, USA. [3] Department of Molecular Biology, Radboud Institute of Molecular Life Sciences and Faculty of Science, Radboud University, Nijmegen, 6525 Gelderland, The Netherlands. [4] Samaritan Medical Center, 830 Washington Street, Watertown, NY 13601, USA. [5] College of Engineering and Applied Science, University of Cincinnati, Cincinnati, OH 45221, USA. [6] Division of Liver Surgery, The Third Affiliated Hospital of Sun Yat-sen University, Guangzhou, Guangdong 510630, China. [7] Department of Environmental Health, University of Cincinnati College of Medicine, Cincinnati, OH 45267, USA. [8] Division of Cardiovascular Health and Disease, Department of Internal Medicine, Heart, Lung and Vascular Institute, University of Cincinnati College of Medicine, Cincinnati, OH 45267, USA. [9] Department of Molecular Genetics, Biochemistry, and Microbiology, University of Cincinnati College of Medicine, Cincinnati, OH 45267, USA. Wei Huang, Yuliang Feng and Jialiang Liang contributed equally to this work. Correspondence and requests for materials should be addressed to X.-Y.Y. (email: yuxycn@aliyun.com) or to Y.W. (email: yi-gang.wang@uc.edu)

The adult human heart fails to replenish the massive loss of cardiomyocytes (CMs) caused by ischemia, which is the leading cause of death worldwide[1]. Intensive research has recently focused on the development of regenerative therapies for ischemic heart disease. Current regenerative approaches are designed to repopulate lost CMs through transplantation of exogenous stem cells from various sources with committed cardiogenic potential[2–4]. The inability to differentiate efficiently, poor cell survival, immaturity of differentiated CM, and arrhythmia have all hampered the application of stem cell-based therapy in clinical settings[5]. Alternatively, cell-free approaches (such as stimulation of endogenous CM proliferation) have emerged as an attractive option for promoting myocardial regeneration.

There has been a longstanding dogma that adult mammalian CMs are incapable of cell division. Recent studies, however, have shown that 1-day-old neonatal mouse can regenerate its heart through dedifferentiation and proliferation of pre-existing CMs[6,7], a phenomenon that is observed in lower vertebrates such as adult zebrafish and amphibians[8]. Unlike the adult zebrafish, the capacity of the neonatal mouse heart to regenerate is diminished as early as 1 week after birth and remains limited throughout adulthood. Various hypotheses have been proposed to explain the varying capacities of different species to undergo cardiac regeneration. Recent compelling evidence showing that CMs have the potential to divide implies there is a latent regenerative potential when endogenous CMs are triggered to proliferate. The most exciting one is the finding of limited self-renewal of human adult CMs using measurements of carbon-14 ($^{14}C$) content by accelerator mass spectrometry[9–11]. The reactivation of CM proliferation, therefore, becomes even more appealing for the potential of heart regeneration. Whether, how and to what extent the endogenous proliferative ability of CMs is sufficient to restore adult heart function, however, remains largely unknown.

MicroRNAs (miRNAs) constitute a class of small noncoding RNAs that bind to the 3′ (untranslated region) UTR of target mRNAs, resulting in the reduction of protein expression predominantly by destabilizing the target mRNAs and/or by inhibiting translation[12,13]. The miRNAs play pivotal roles in many biological processes including apoptosis of CMs during MI[14]. However, the miRNAs that regulate CM proliferation during homeostasis and injury are not fully defined.

In this study, we first show that the expression of cardiac miR-128 is lower in neonates than in adults, and is reduced during neonatal heart regeneration. Furthermore, cardiac-specific overexpression of miR-128 in early postnatal mice suppresses CM proliferation and causes impaired cardiac function. Conversely, knockout of miR-128 reactivates CM proliferation and cardiac regeneration in the adult mice, in part through modulation of cell cycle-related genes by targeting Suz12 in the heart. Collectively, our results suggest that miR-128 functions as a critical regulator of endogenous cardiac proliferation and regeneration.

## Results

**MiR-128 increases during postnatal heart growth.** RNA sequencing (RNA-seq) in mouse cardiac ventricles was performed on postnatal days 1, 7, and 28 (P1, P7, and P28) to identify potential miRNAs involved in the regulation of postnatal heart growth. MiR-128 was robustly upregulated in P7 hearts as compared to P1, which was further confirmed by quantitative PCR (qPCR) array (Supplementary Fig. 1A). As previously reported[15], miR-128 was predominantly expressed in brain tissue but was also expressed in the heart (Supplementary Fig. 1B). Its expression in adult myocardium was further confirmed by in situ

hybridization (ISH) (Supplementary Fig. 1C). To investigate the role of miR-128 in cell cycle withdrawal during heart growth, mouse hearts were harvested and sectioned at P1, P7, and P28 (Fig. 1a). As neonates (P1) progress to adulthood (P28), CMs underwent a maturation process characterized by suppression of cell proliferation as evidenced by decreased numbers of Ki67+ CMs (Fig. 1a, b). In addition, cardiac mass increased from P1 to P28 primarily due to an increase in CM size rather than in number (Fig. 1c, d). Interestingly, we found that miR-128 expression was significantly increased during heart development (Fig. 1e). Furthermore, the level of miR-128 was found to be significantly elevated in P7 and P28 hearts compared with P1 hearts. In order to examine whether the postnatal upregulation of miR-128 occurs specifically in the CMs, we isolated CMs from P1 and P28 hearts, respectively, and found significantly higher expression of miR-128 in P28 CMs when compared with P1 CMs (Fig. 1f). Moreover, the expression of miR-128 in CMs was significantly higher than in non-CMs (e.g., cardiac fibroblasts, CFs) (Fig. 1g). These data indicate a potential role for miR-128 in regulating CM proliferation.

**Overexpression of miR-128 impairs cardiac homeostasis.** To specify the function of miR-128 in the heart, a mouse model was generated in which miR-128 expression was under control of the α-myosin heavy chain (α-MHC) promoter that was under temporal regulation by doxycycline (Dox). This "Tet-off" transgenic mouse (α-MHC-tTA; miR-128$^{TetRE}$) was produced by crossing α-MHC-tTA mice with miR-128$^{TetRE}$ mice (Supplementary Fig. 2A). In α-MHC-tTA; miR-128$^{TetRE}$ mice, the TetRE portion of tTA can bind to the TetO sequences after Dox withdrawal, and subsequently induce the CM-specific overexpression of miR-128 (designated as miR-128$^{OE}$ mice) in defined temporal windows (Fig. 2a and Supplementary Fig. 2B). In general, induced transgene expression begins during the second week of Dox withdrawal due to the slow clearance of Dox from tissues[16–18]. Withdrawal of Dox from miR-128$^{OE}$ fetuses starting at embryonic day 6 (E6) resulted in significant induction of miR-128 in miR-128$^{OE}$ hearts at the P1 neonatal stage as determined by qPCR (Fig. 2b). At P1, the explanted hearts from miR-128$^{OE}$ mice were markedly enlarged (Fig. 2c) compared with hearts from miR-128$^{TetRE}$ mice (Control mice, designated as Ctrl). The higher heart-to-body weight ratios (HW/BW) of miR-128$^{OE}$ mice relative to Ctrl showed a progressive increase in heart mass (Fig. 2d). Morphologically, CM size was measured by staining with wheat germ agglutinin (WGA), which showed that miR-128$^{OE}$ CMs were significantly larger than Ctrl CMs (Fig. 2e, f), implying the development of cardiac hypertrophy. Cardiac function analysis from miR-128$^{OE}$ mice at P1 by echocardiography (Fig. 2g) showed a reduction of left ventricular (LV) ejection fraction (EF) and fractional shortening (FS), parameters of cardiac contractile function, when compared with Ctrl mice. In contrast, Dox treatment did not induce cardiac dysfunction in Ctrl mice (Supplementary Fig. 2C).

To explore the cellular mechanisms underlying the observed hypertrophy, heart sections were immunostained to assess proliferation and apoptosis. The miR-128$^{OE}$ hearts displayed diminished proliferation of CMs based on the reduced number of Ki67+ CMs compared with Ctrl (Fig. 2h, i). However, there was no significant increase in apoptotic CMs in miR-128$^{OE}$ hearts when assessed by TUNEL staining (Supplementary Fig. 2D).

To study the role of miR-128 in heart development, miR-128$^{OE}$ mice were mated in the absence of Dox (Supplementary Fig. 2E). Assessment of miR-128 level by qPCR confirmed its marked overexpression by E10.5 in the hearts of miR-128$^{OE}$ mice (Supplementary Fig. 2F). These miR-128$^{OE}$ mutant mice

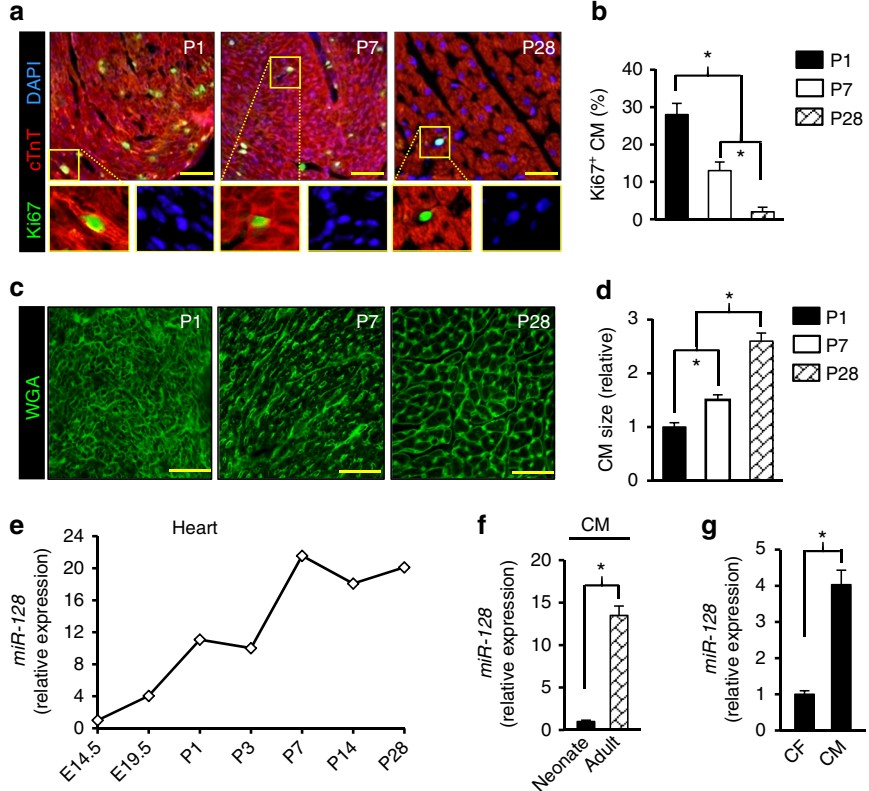

**Fig. 1** *MiR-128* increases as the heart progresses from neonate to adult. **a** Evaluation of wild-type mouse cardiomyocyte (CM) proliferative activity using Ki67 immunostaining at postnatal day 1 (P1), P7, and P28. Scale bars, 25 μm. **b** Percentage of CM Ki67$^+$/cTnT$^+$ in P1, P7, and P28 hearts ($n = 5$ mice for each time point, ~600 CMs/heart). **c** Wheat germ agglutinin (WGA) staining of P1, P7, and P28 hearts. Scale bars, 25 μm. **d** Quantification of CM size in P1, P7, and P28 hearts stained with WGA ($n = 5$ mice for each time point, ~250 CMs/heart). **e** Evaluation of *miR-128* expression level during heart development using qPCR analysis, including embryonic day 14.5 (E14.5), E19.5, P1, P3, P7, P14, and P28 hearts ($n = 5$). **f** qPCR analysis of *miR-128* expression in neonatal (P1) and adult (P28) CMs ($n = 5$). **g** Comparison of *miR-128* expression by qPCR in cardiac fibroblast (CF) and CMs ($n = 5$). Statistical significance was calculated using ANOVA in **b**, **d** and Student's *t*-test in **f**, **g**. Data are represented as means ± SEM. *$P < 0.05$

displayed enlarged heart chambers, myocardial fibrosis, CM hypertrophy, and impaired LV systolic heart function at P28 (Supplementary Fig. 2G–I). Moreover, KEGG pathway analysis showed that oxidative phosphorylation, metabolism, hypertrophic cardiomyopathy, and dilated cardiomyopathy pathways were enriched in *miR-128*$^{OE}$ hearts. Concomitantly, cell cycle and DNA replication pathways were suppressed in *miR-128*$^{OE}$ hearts (Supplementary Fig. 3). Taken together, these data indicate that CM-specific overexpression of *miR-128* induces early CM cell cycle exit, compensatory pathological growth of CM (hypertrophy), and impaired cardiac homeostasis.

**Deletion of *miR-128* stimulates postnatal CM proliferation**. Having established a correlation between *miR-128* overexpression and inhibition of CM proliferation, we asked whether loss of *miR-128* is causal for CM proliferation. In vitro when *miR-128* was knocked down using a specific *miR-128* inhibitor (designated as Anti-*miR-128*) (Supplementary Fig. 4A), the neonatal CMs became dedifferentiated after 7 days. Loss of the CM differentiated state was based on sarcomere disassembly[4] assessed by immunostaining for cardiac troponin T (cTnT), a marker for sarcomere integrity (Supplementary Fig. 4B–C). Consistent with sarcomere disassembly, expression of sarcomere genes (*Tnnt2* and *Myh6*) was reduced (Supplementary Fig. 4D). While promoting loss of differentiation, silencing of *miR-128*, did not induce apoptosis in these cells (Supplementary Fig. 4E).

The effect of *miR-128* knockdown on CM proliferation was then examined using phosphorylated histone 3 (pH3, a marker of

mitosis) and Aurora B kinase (a marker of cytokinesis). In addition to inducing dedifferentiation, silencing of *miR-128* (anti-*miR-128* CMs) increased the number of mitotic CMs compared with control CMs (Ctrl) as determined by immunostaining for pH3 (Supplementary Fig. 4F). Expression of Aurora B kinase was markedly elevated in Anti-*miR-128* CMs (Supplementary Fig. 4G). We also found a significant increase in the number of 5-ethynyl-2´-deoxyuridine (EdU) positive CMs in the Anti-*miR-128* group indicative of elevated DNA replication (Supplementary Fig. 4H). Importantly, an increased level of GATA4 (a marker for dedifferentiated CMs[8]), was observed in Anti-*miR-128* CMs (Supplementary Fig. 4I).

Given the evidence that silencing of *miR-128* induces CM proliferation in vitro, we next preceded to determine the effects that deletion of *miR-128* would have on CM proliferation in vivo. Cardiac-specific conditional *miR-128* knockout mice were generated by crossing *miR-128*$^{flox/flox}$ (*miR-128*$^{fl/fl}$) mice (Supplementary Fig. 5A) with *Nkx2.5*$^{Cre}$ mice, resulting in cardiac-specific deletion of *miR-128* during cardiogenesis (*Nkx2.5*$^{Cre}$; *miR-128*$^{fl/fl}$ mice, designated as *miR-128*$^{-/-}$) (Supplementary Fig. 5B). Hearts from *miR-128*$^{fl/fl}$ (Control mice, Ctrl) and *miR-128*$^{-/-}$ mice were harvested and analyzed at P7, at the time when most CMs have exited the cell cycle and become post-mitotic[6,7]. Downregulation of *miR-128* in hearts from *miR-128*$^{-/-}$ mice was confirmed by qPCR (Fig. 3a, b). By E10.5, *miR-128*$^{-/-}$ hearts exhibited marked downregulation of *miR-128*. Phenotypic characterization of *miR-128*$^{-/-}$ mice at P7 demonstrated that heart size (Fig. 3c) and heart function by echocardiography

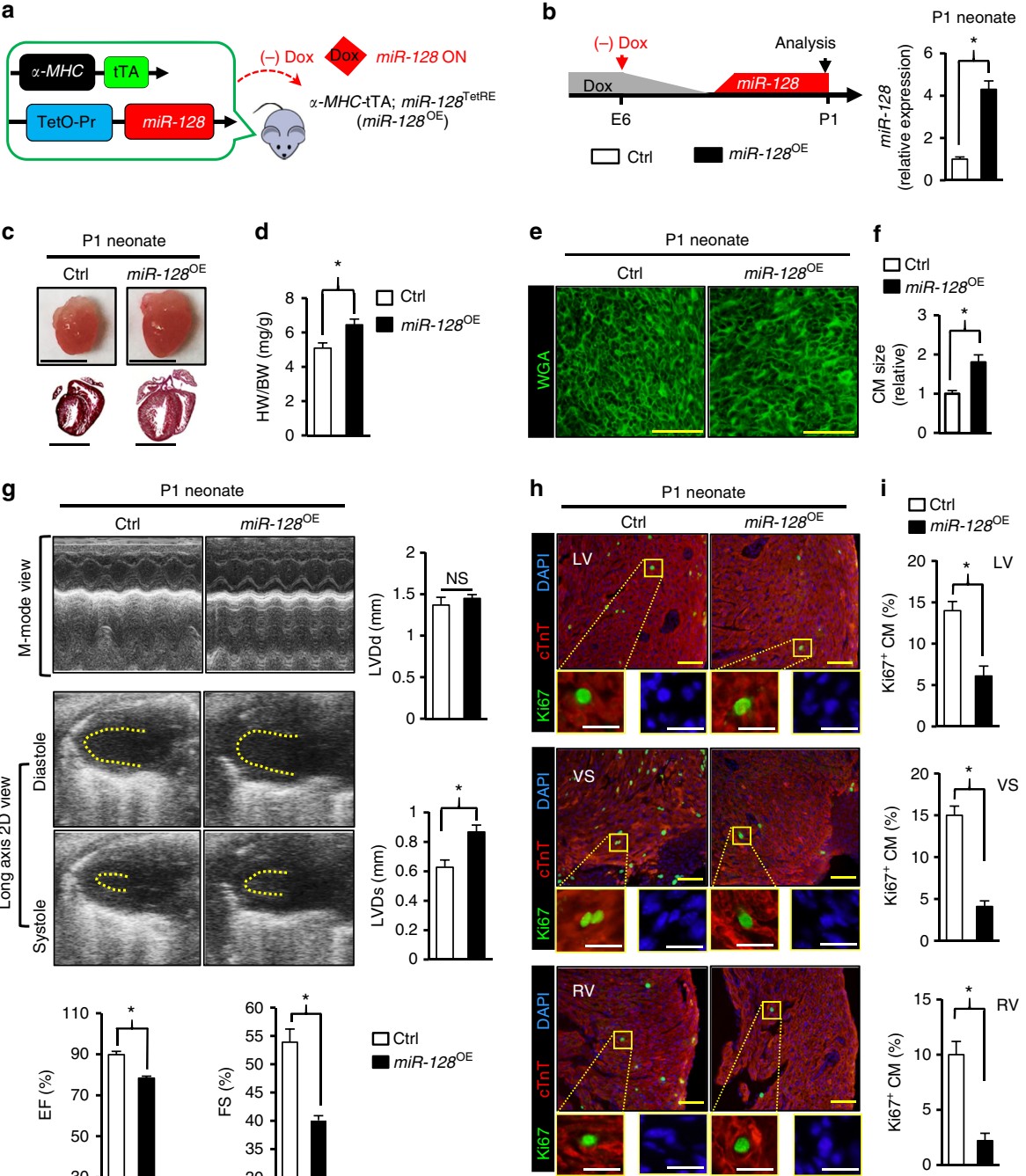

**Fig. 2** Overexpression of *miR-128* in cardiomyocytes impairs cardiac homeostasis. **a** Schematic showing the generation of mice that overexpress CM-specific *miR-128* after doxycycline (Dox) withdrawal. Control mice were *miR-128*^TetRE mice, and *miR-128*^OE mice were *α-MHC*-tTA; *miR-128*^TetRE mice. **b** Schematic of experimental design for CM-specific overexpression of *miR-128* at P1 (left panel). Right panel shows the qPCR analysis of *miR-128* expression in Ctrl and *miR-128*^OE mice (n = 5). **c** Gross morphology (upper panel) and Masson trichrome staining (lower panel) of hearts at P1. Scale bars, 0.25 cm (upper panel); 1 mm (lower panel). **d** Heart weight (HW) to body weight (BW) ratio of P1 mice (n = 8). **e** Wheat germ agglutinin (WGA) staining of P1 neonatal hearts. Scale bars, 25 μm. **f** Quantification of CM size as determined by WGA staining (n = 8 mice, ~200 CMs/heart). **g** Heart function analyzed by echocardiography in P1 mice as measured by left ventricular end-diastolic diameter (LVDd), LV end-systolic diameter (LVDs), ejection fraction (EF), and fraction shortening (FS) (n = 6). **h** Evaluation of CM proliferative activity by Ki67 immunostaining in Ctrl and *miR-128*^OE hearts. Scale bars, 25 μm (yellow); 10 μm (white). **i** Quantification data of CM proliferative activity by Ki67 staining in Ctrl and *miR-128*^OE hearts (n = 6 mice, ~800 CMs/heart). Statistical significance was calculated using Student's *t*-test. Data are represented as means ± SEM. *P < 0.05

(Fig. 3d) were unaffected by *miR-128* deletion. Although the heart weight-to-body weight ratio (HB/WB) of *miR-128*^−/− and Ctrl mice at P7 was similar (Fig. 3e), the CMs in *miR-128*^−/− hearts were smaller (Fig. 3f). This could indicate an increased number of CMs in these hearts due to persistent proliferation resulting from

*miR-128* deletion. To test this proposition, CMs were stained with Ki67 to assess the number of cycling cells. The results showed that loss of *miR-128* resulted in a striking increase in CM proliferation (Fig. 3g, h). Sarcomere disassembly, a characteristic of CM dedifferentiation and proliferation, was also prominent in *miR-*

$128^{-/-}$ hearts compared with the Ctrl hearts (Fig. 3g, i, and j). In addition to the CMs with disassembled sarcomeres, there was a significantly higher number of Ki67-positive cells in the $miR\text{-}128^{-/-}$ hearts, and with no obvious CM apoptosis in the $miR\text{-}128^{-/-}$ hearts (Fig. 3k). By EdU incorporation assay (Fig. 3l–o), we also found a significant increase in the number of EdU$^+$ CMs in $miR\text{-}128^{-/-}$ hearts compared with Ctrl at P14 (Fig. 3l, m) as well as at P21 (Fig. 3n, o). Nevertheless, these mice developed normally to adulthood and did not exhibit any cardiac

dysfunctions (Supplementary Fig. 6A, B). These data suggest that tissue-specific deletion of $miR\text{-}128$ deletion is sufficient to extend the postnatal CM proliferation window.

**$MiR\text{-}128$ deletion reconfigures cell cycle gene expression.** RNA-seq was performed on control (Ctrl) and $miR\text{-}128^{OE}$ hearts (P7) to identify the putative target genes of $miR\text{-}128$ responsible for cell cycle regulation. By comparing the downregulated mRNAs identified in $miR\text{-}128^{OE}$ hearts relative to Ctrl hearts with all

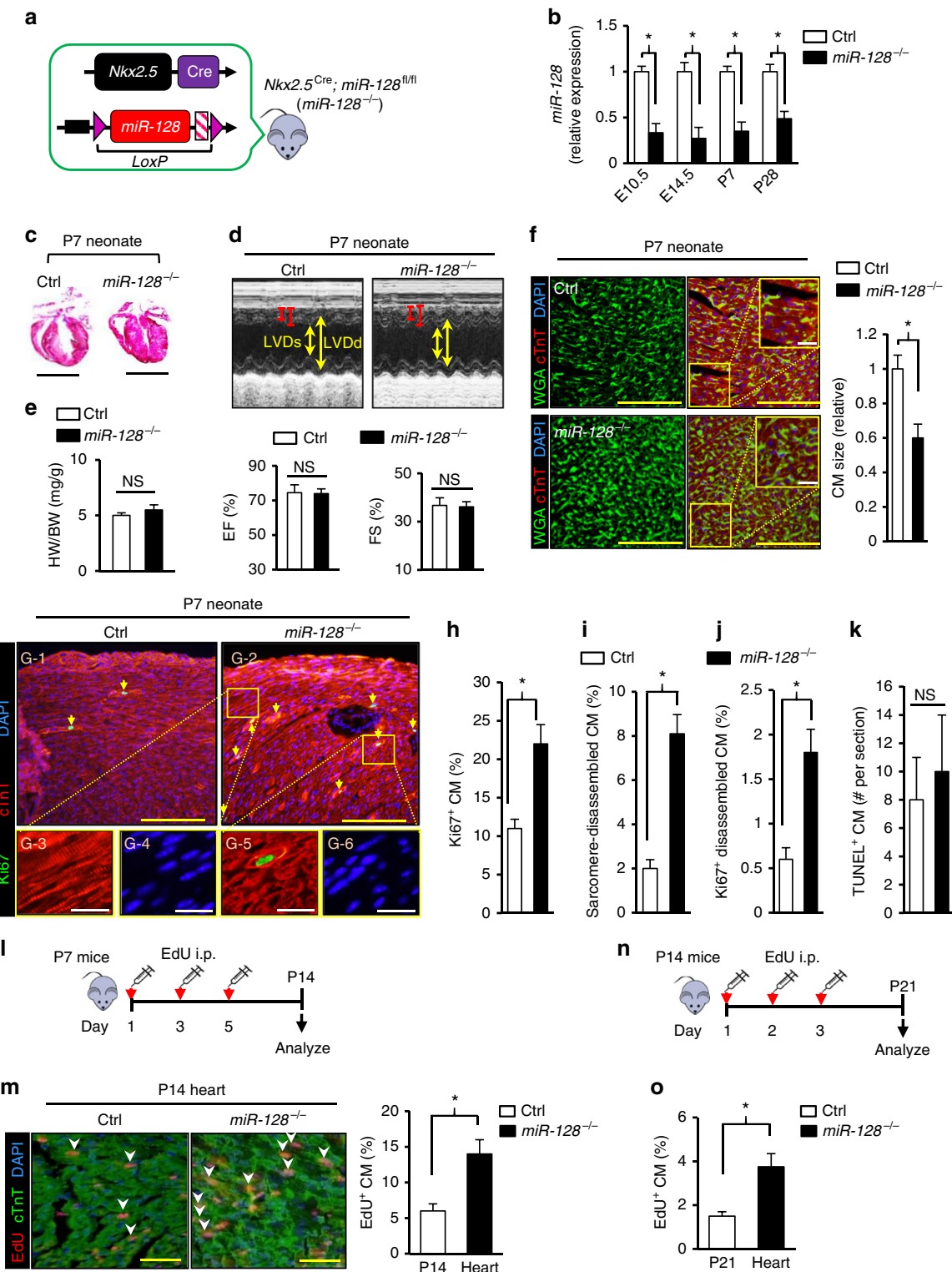

possible predicted candidate *miR-128* target genes[19], we found 87 genes that contained the predicted binding site at the 3′UTR (Supplementary Fig. 7A). Gene Ontology (GO) PANTHER Analysis was then performed to identify the affected cellular biological processes. The leading biological category was "cellular process" category, with nearly 28.7% of all associated genes (GO: 0009987, Supplementary Fig. 7B). A further subgroup analysis of the "cellular process" indicated the potential for *miR-128* to affect multiple pathways that are related to regulation of the cell cycle, cell communication, and cellular component movement (Supplementary Fig. 7B). Moreover, the analysis of genes down-regulated in *miR-128*^OE showed statistically significant enrichment of genes downregulated after small interfering RNA (siRNA) inhibition of components of polycomb repressive complex 2 (PRC2), *Suz12* in particular (Supplementary Fig. 7C). PRC2 is a chromatin modifier complex that is crucial for organogenesis[20]. Perturbation of the epigenetic landscape by ablation of PRC2 subunits during early cardiac development inhibits CM proliferation, and eventually leads to fatal cardiac malformations[21]. Computational analysis showed that *Suz12* was a predicted target gene of *miR-128* (Fig. 4a). In contrast to neonatal hearts, the protein levels of SUZ12 were lower in the adult heart (where the CM proliferation ability is quite limited) (Fig. 4b, c), paralleling the upregulation of *miR-128*. These data suggest that *miR-128* may regulate CM proliferation via its intercation with *Suz12* .

To investigate whether *miR-128* regulates *Suz12* expression, mouse neonatal CMs were transfected with a negative control (Ctrl), a mimic of *miR-128* (*miR-128*), or an inhibitor of *miR-128* (Anti-*miR-128*) and assessed for the level of SUZ12 by western blot analysis. Overexpression of *miR-128* significantly reduced the protein level of SUZ12, whereas inhibition of *miR-128* led to its increased expression (Fig. 4d, e). To further test whether *miR-128* regulates *Suz12* expression, we constructed a vector containing luciferase reporter with a DNA sequence encoding the complete 3′UTR from mouse *Suz12* (designated as WT), and a mutated vector (designated as Mut) containing mismatches in the predicted *miR-128*-binding site in the 3′UTR (Fig. 4f). Co-transfection of HEK293T cells with the *Suz12* 3′UTR plasmid (WT) and *miR-128* mimic resulted in a significant decrease in luciferase activity compared with cells co-transfected with the negative control or the mutated 3′UTR target sequence (Mut), indicating that *Suz12* is a direct target of *miR-128*, consistent with a previous report[22].

To better define how the interaction between *miR-128* and *Suz12* might mediate cell proliferation in vivo, we first analyzed the expression of cell cycle-related genes in *miR-128*^−/− hearts at P7 and found that expression of SUZ12, cyclin E and cyclin-dependent Kinase 2 (CDK2) was elevated in *miR-128*^−/− P7 hearts compared with hearts from control mice (Ctrl) while the CDK inhibitor (CDKi) p27 was downregulated (Fig. 4g, h). As SUZ12

can modulates the functionality of enhancer of zeste homolog 2 (EZH2), which catalyzes the formation of H3K27me3 (a transcriptional repressive mark)[23], we performed chromatin immunoprecipitation (ChIP)-qPCR assay and found that SUZ12, EZH2, and H3K27me3 were significantly enriched on the *p27* promoter in *miR-128*^−/− hearts as compared to Ctrl (Fig. 4i). These data indicate that the downregulation of *p27* induced by *miR-128* deletion is attributed, at least in part, to PRC2 mediated gene silencing.

To investigate whether regulation of *Suz12* signaling after *miR-128* deletion is responsible for the enhanced proliferation in CMs, loss of function study was performed using siRNA against *Suz12*. In vitro, direct inhibition of *Suz12* by siRNA (si-*Suz12*) in cultured *miR-128*^−/− neonatal CMs reversed the pro-proliferative effect conferred by *miR-128* deletion (*miR-128*^−/−), as evidenced by a significant decrease in the number of Ki67^+ CMs in the si-*Suz12* group in contrast to control group (si-Ctrl) (Fig. 5a–c). To further validate that the *Suz12*-pathway is a major functional mediator of *miR-128* effects, we injected *miR-128*^−/− mice (intraperitoneal (i.p.) injection) with si-*Suz12* or si-Ctrl at P1, P3, P5, and harvested the hearts at P7 (Fig. 5d). Knockdown of *Suz12* in vivo significantly induced CM hypertrophy (Fig. 5e) and impaired CM proliferation by decreasing the number of EdU^+ CMs (Fig. 5f). Moreover, there was a significant increase in the level of p27 and decrease of Cyclin E and CDK2 expression in the si-*Suz12* group when compared with si-Ctrl treated hearts (Fig. 5g, h). These data indicate that *miR-128* deletion stimulates proliferation of CMs, in part through epigenetic modulation of cell cycle-related genes via targeting of *Suz12* (Fig. 5i).

**Overexpression of *miR-128* inhibits cardiac regeneration.** An apex resection (AR) model in neonatal mice at P1 (Supplementary Fig. 8A) was developed to enable assessment of temporal gene expression during cardiac regeneration. Histological analysis verified that by day 7 post AR, the initial large blood clot in the apex had been replaced by newly formed CMs and limited fibrotic tissue (Supplementary Fig. 8B). Also at day 7 post AR, genes associated with cell proliferation were significantly activated, whereas *miR-128* expression was significantly diminished (Supplementary Fig. 8C, D). These data imply that expression level of *miR-128* is associated with neonatal heart regeneration.

A *miR-128*^OE mouse model in which *miR-128* was over-expressed in a CM-specific and temporally controlled (by Dox withdrawal) manner (Fig. 6a) was used to test whether *miR-128* regulates cardiac regenerative capacity in neonatal mice. The *miR-128*^OE mice and control *miR-128*^TetRE mice (Ctrl) were subjected to AR at P1, and hearts from both groups were examined histologically. At 21 days post AR, the *miR-128*^OE hearts showed left ventricle dilation and defective regeneration compared with Ctrl groups (Fig. 6b). The *miR-128*^OE hearts

**Fig. 3** Cardiac *miR-128* deletion promotes postnatal CM proliferation in vivo. **a** Schematic diagram depicting the generation of cardiac-specific *miR-128* knockout (*miR-128*^−/−) mice. Control mice were *miR-128*^fl/fl mice, and *miR-128*^−/− mice were Nkx2.5^Cre; *miR-128*^fl/fl mice. **b** The expression level of *miR-128* during heart development (n = 6) analyzed by qPCR, including embryonic day 10.5 (E10.5), E14.5, postnatal day 7 (P7), and P28. **c** Masson trichrome staining of mouse hearts at P7. Scale bars, 2.0 mm. **d** Comparison of cardiac function between Ctrl and *miR-128*^−/− hearts analyzed by echocardiography at P7, and measured by EF and FS (n = 6). **e** Measurement of HW to BW ratio in Ctrl and *miR-128*^−/− mice (n = 6). **f** Evaluation of CM size in P7 Ctrl and *miR-128*^−/− hearts assessed by WGA and cardiac troponin T (cTnT) staining (n = 5 mice, ~250 CMs/heart). Scale bars, 50 μm (yellow); 10 μm (white). **g** Assessment of CM proliferative activity and sarcomere structure in P7 hearts by immunofluorescence of cTnT and Ki67. Arrows indicate Ki67-positive CMs with sarcomere disassembly. Scale bars, 500 μm (yellow); 25 μm (white). **h–j** Quantification of Ki67^+ CMs, sarcomere disassembled CMs and Ki67^+ disassembled CMs (n = 4860 CMs pooled from six mice). **k** CM apoptosis analyzed by TUNEL staining. **l** Schematic diagram depicting the protocol for EdU intraperitoneal (i.p.) injection at P7 mice to label proliferating CMs in vivo. **m** Analysis of CM proliferation by EdU incorporation assay in Ctrl and *miR-128*^−/− hearts at P14 (n = 6 mice, ~250 CMs/heart). Scale bars, 50 μm. **n** Schematic diagram depicting the protocol for EdU intraperitoneal (i.p.) injection at P14 to label proliferating CMs in vivo. **o** Comparison of EdU^+ CMs in Ctrl and *miR-128*^−/− hearts at P21 (n = 6 mice, ~200 CMs/heart). Statistical significance was calculated using Student's *t*-test. Data are represented as means ± SEM. *P < 0.05

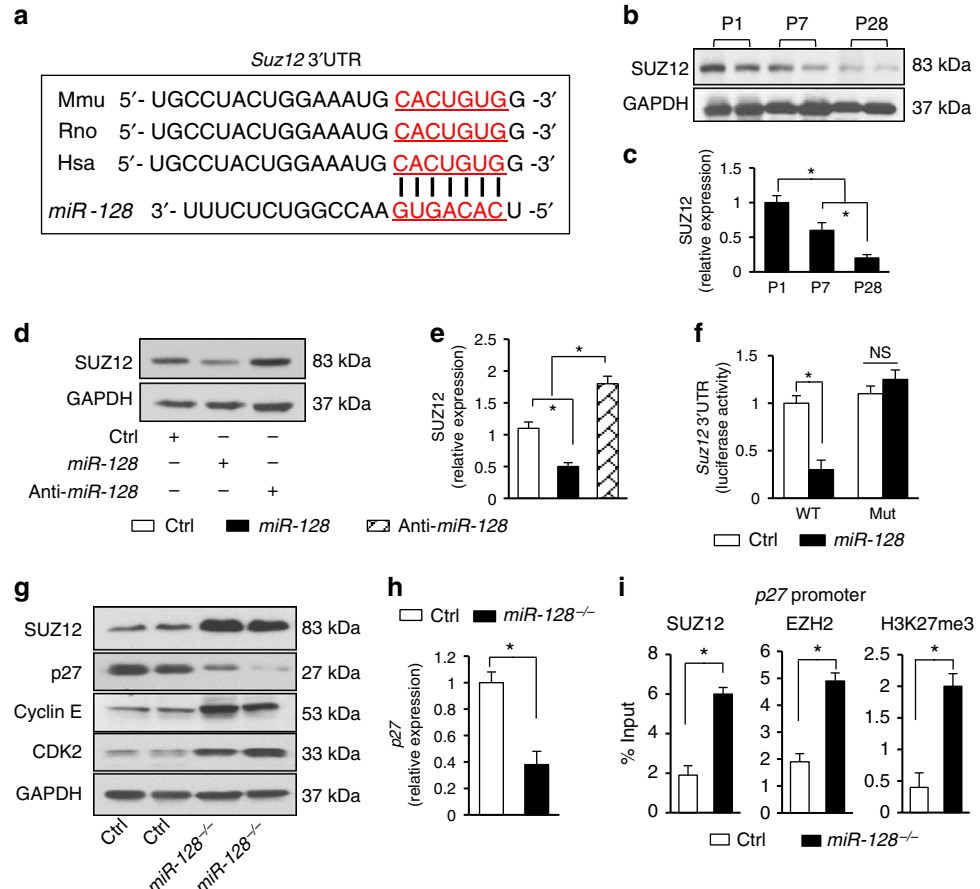

**Fig. 4** *MiR-128* deletion is associated with activation of cell cycle-related genes. **a** The predicted conserved target site of *miR-128* in the 3′UTR of *Suz12* from different species. **b**, **c** Western blot analysis of SUZ12 expression in mouse hearts at P1, P7, and P28 ($n = 5$). **d**, **e** Western blot analysis of SUZ12 expression in neonatal CMs treated with either vehicle (Ctrl), *miR-128* mimic (*miR-128*) or *miR-128* inhibitor (Anti-*miR-128*) ($n = 3$). **f** Luciferase reporter assay for wild-type (WT) and mutant *Suz12* 3′UTR (Mut) in cells treated with either vehicle (Ctrl) or *miR-128* mimic (*miR-128*) ($n = 3$). **g** Western blot assay for cell cycle-related protein expression in control (*miR-128*^fl/fl), and *miR-128*^−/− (*Nkx2.5*^Cre; *miR-128*^fl/fl) hearts at P7 ($n = 5$). **h** Quantification data of *p27* mRNA levels in Ctrl and *miR-128*^−/− hearts by qPCR ($n = 5$). **i** Comparison of SUZ12, EZH2, and H3K27me3 enrichment on the *p27* promoter by ChIP-qPCR ($n = 5$). Statistical significance was calculated using ANOVA in **c**, **e** and Student's *t*-test in **f**, **h**, and **i**. Data are represented as means ± SEM. *$P < 0.05$. NS designates not significant

showed fewer proliferating CMs, as quantified by the decreased number of EdU$^+$ CMs in the injured apex and border area (Fig. 6c, d), and a greater extent of CM hypertrophy (Fig. 6e). In addition, systolic function was significantly impaired in the *miR-128*^OE group relative to Ctrl group (Fig. 6f, g). These findings suggest that inhibition of CM proliferation by *miR-128* over-expression can impair cardiac regeneration in a neonatal mouse model.

**Deletion of *miR-128* promotes adult cardiac regeneration.** To investigate whether loss of *miR-128* in the adult is capable of promoting CM proliferation, cardiac-specific, tamoxifen (TAM) inducible *miR-128* knockout mice were then generated by crossing α-MHC^MerCreMer mice with *miR-128*^fl/fl mice (Fig. 7a). TAM was administered at P21 to induce the *miR-128* knockout at the adult stage. The adult *miR-128* deleted mice were designated iKO, and the knockout was validated by qPCR. The heart weight-to-body ratio (HB/WB) was unchanged in iKO mice. Staining with WAG, however, showed that the size of the iKO CMs was smaller than the control CMs (Fig. 7b–d), suggesting that loss of *miR-128* in the adult heart increases the number of CMs. This increase in CM number following *miR-128* deletion was further confirmed by analysis of EdU incorporation into CMs (Fig. 7e).

Furthermore, the total number of adult CMs and percentage of mono-nucleated CMs was significantly increased in iKO hearts 2 weeks after TAM-induced *miR-128* deletion (Fig. 7f, g).

To determine whether cells in the myocardial lineage dedifferentiate following deletion of *miR-128*, a TAM inducible dual-lineage tracing system was generated by crossing α-MHC^MerCreMer mice with *miR-128*^fl/fl mice followed by crossing with Rosa26-tdTomato reporter mice to produce α-MHC^MerCreMer; *miR-128*^fl/+; R26R-tdTomato mice (designated as iKO-tdTomato) (Fig. 7h). In these transgenic mice, the pre-existing CMs with *miR-128* knockout were labeled red (tdTomato, red fluorescence) following TAM administration. After TAM-induced *miR-128* deletion (Fig. 7i), the α-MHC myocardial lineage-positive CMs in iKO-tdTomato mouse displayed a disorganized sarcomere structure and reduced sarcomere-related gene expression compared with control mice (α-MHC^MerCreMer; R26R-tdTomato, designated as Ctrl-tdTomato) (Fig. 7j, k). There was no observable apoptosis in hearts from iKO-tdTomato mice (Supplementary Fig. 9A, B). There was, however, increased expression of genes associated with cell proliferation (*Nuspa1*, *Racgap1* and *Myh10*) and fetal genes associated with negative regulation of CM differentiation (*Nppa* and *Nppb*) in iKO-tdTomato hearts detected by qPCR (Fig. 7k). Importantly, in iKO mice, cardiac morphology remained

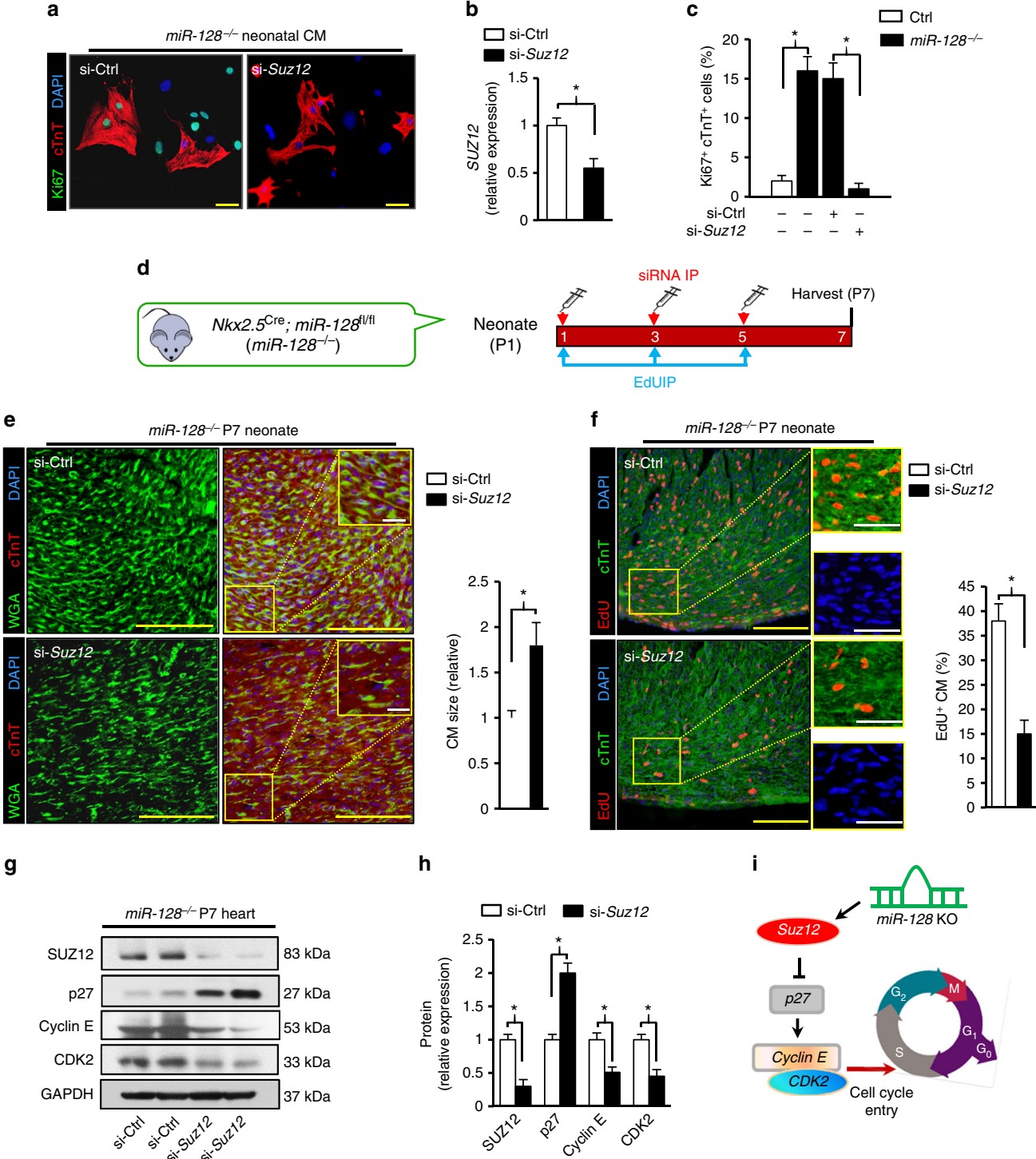

**Fig. 5** *MiR-128* regulates CM proliferation through targeting *Suz12*. **a** In vitro, evaluation of CM proliferation by immunofluorescence staining of Ki67 in *miR-128^-/-* (*Nkx2.5^Cre; miR-128^fl/fl*) neonatal CMs transfected with either scrambled control siRNA (si-Ctrl) or *Suz12* siRNA (si-Suz12). Cells are counter-stained with DAPI to visualize nuclei and with antibody to cTnT to identify CMs. Scale bars, 40 μm. **b** Expression of *Suz12* in *miR-128^-/-* neonatal CMs transfected with either a scrambled control siRNA (si-Ctrl) or *Suz12* siRNA (si-Suz12) (*n* = 5). **c** Quantification of CM proliferation by Ki67 immunostaining (*n* = 12 samples, ~150 CMs/sample). **d** Schematic diagram depicting the protocol for siRNA and EdU intraperitoneal (i.p.) injection for P1 mice. **e** CM size analysis by WGA and cTnT staining in si-Ctrl and si-Suz12 treated *miR-128^-/-* hearts at P7 (*n* = 5 mice, ~300 CMs/heart). Scale bars, 50 μm (yellow); 10 μm (white). **f** Comparison of EdU^+ CMs in si-Ctrl and si-Suz12-treated *miR-128^-/-* hearts at P7 (*n* = 5 mice, ~400 CMs/heart).). Scale bars, 50 μm (yellow), 20 μm (white). **g, h** Western blot analysis of cell cycle-related genes in si-Ctrl and si-Suz12 treated *miR-128^-/-* hearts at P7 (*n* = 3). **i** Proposed model by which *miR-128* deletion promotes CM proliferation through coordinating the expression of cell cycle-related genes. Statistical significance was calculated using Student's *t*-test in **b**, **c**, **e**, **f**, and **h**. Data are represented as means ± SEM. *$P < 0.05$

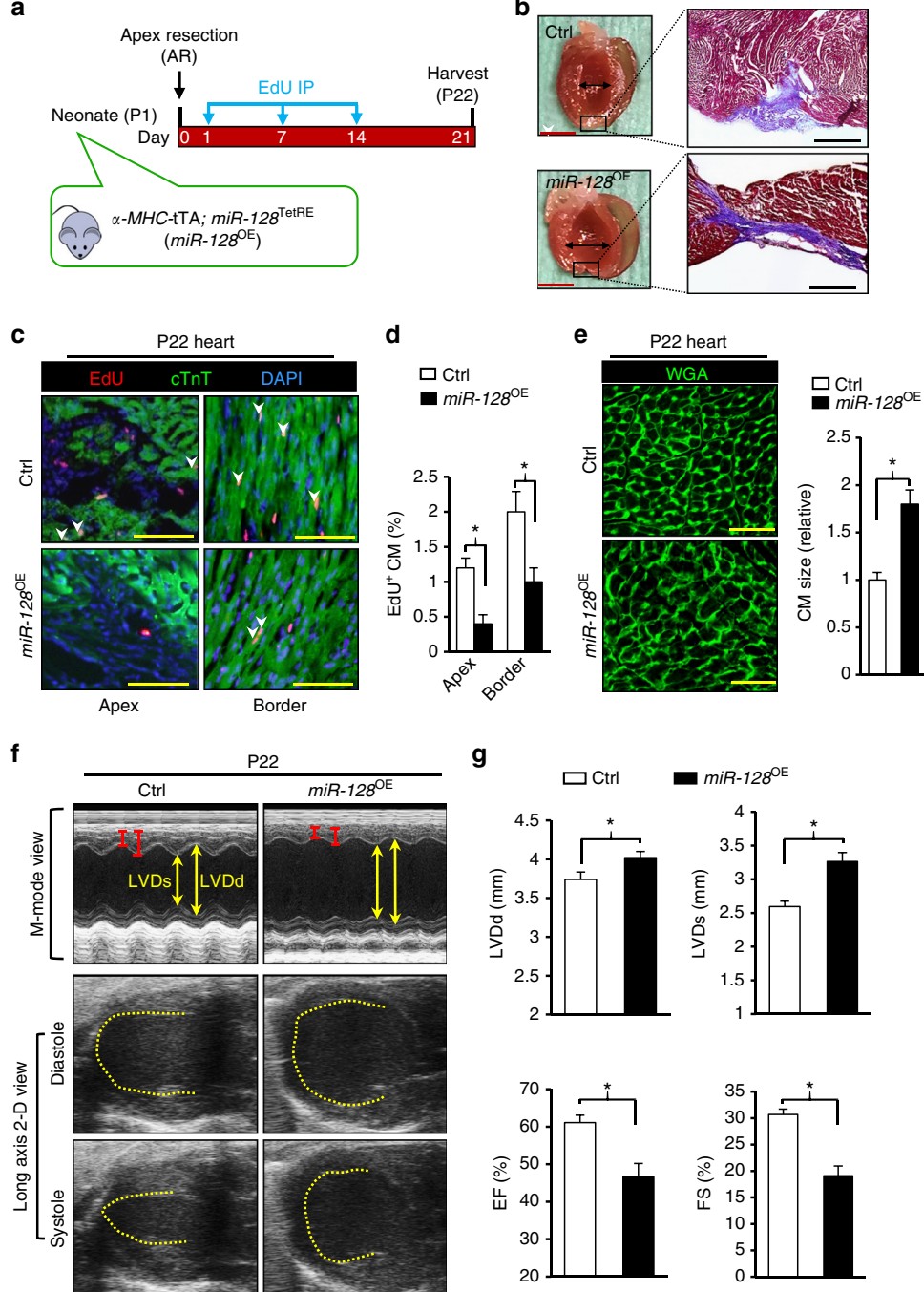

**Fig. 6** Overexpression of *miR-128* inhibits neonatal cardiac regeneration. **a** Schematic diagram depicting the timing of apex resection (AR) and EdU labeling for P1 mice. Control (Ctrl) mice were *miR-128*^TetRE mice, and *miR-128* overexpressing mice (*miR-128*^OE) were *α-MHC*-tTA; *miR-128*^TetRE mice. **b** Masson trichrome staining of Ctrl and *miR-128*^OE hearts at day 21 after AR. Scale bars, 0.25 cm (red); 200 μm (black). **c** Evaluation of CM proliferation by EdU incorporation. Scale bars, 50 μm. **d** Quantification of EdU+ CMs in P1 hearts at day 21 post AR ($n = 5$ mice, ~250 CMs/heart). **e** Staining of mouse hearts with WGA at day 21 after AR ($n = 5$ mice, ~200 CMs/heart). Scale bars, 25 μm. **f**, **g** Heart function analyzed by echocardiography and quantification of LVDd, LVDs, EF, and FS ($n = 6$) 21 days post AR. Statistical significance was calculated using Student's *t*-test in **d**, **e**, and **g**. Data are represented as means ± SEM. *$P < 0.05$

unchanged and heart functions were normal (Supplementary Fig. 9C, D). These results indicate that deletion of *miR-128* in the adult heart results in dedifferentiation and cell cycle re-entry of CMs, but has no impact on heart function.

To determine whether induction of cardiac proliferation following *miR-128* deletion in adult mice is sufficient to allow adult heart repair following MI, adult iKO mice were subjected to permanent ligation of the left anterior descending (LAD)

coronary artery. One day after MI, we administered TAM to delete *miR-128* in CMs (Fig. 8a). Analysis of iKO hearts at day 7 after TAM injection showed that the expression of *miR-128* target SUZ12 was significantly increased, accompanied by downregulation of the p27 and upregulation of Cyclin E and CDK2 (Fig. 8b). Moreover, we observed a significant increase in the number of Aurora B-positive CMs (Fig. 8c), and in EdU positive CMs in the iKO hearts (Fig. 8d, e). These changes were

associated with reduced cardiac fibrosis in the iKO hearts (Supplementary Fig. 10B). In addition, analysis of sarcomere structures in iKO hearts revealed robust dedifferentiated cardiac muscle in border areas and remote areas (Fig. 8d). Although it was previously reported that *miR-128* regulates apoptosis by targeting peroxisome proliferator-activated receptor gamma (*Pparg*)[24], we found no significant differences in either PPARγ expression or apoptosis in iKO hearts when compared to Ctrl hearts at day 7 after TAM injection (Supplementary Fig. 10C, D).

In addition to dedifferentiation and enhanced proliferation, iKO hearts showed significantly less fibrosis as compared to Ctrl groups 4 weeks after MI (Fig. 9a, b). Similarly, diminished cardiac function was significantly reversed in iKO mice, as evidenced by increased EF and FS after MI when compared with the Ctrl animals (Fig. 9c, d). Cardiac remodeling was also significantly reversed in iKO mice with reduced LVDd and LVDs (Fig. 9d). Collectively, these data indicate that inhibition of *miR-128* promotes CM proliferation and improves endogenous cardiac regeneration after MI (Fig. 10).

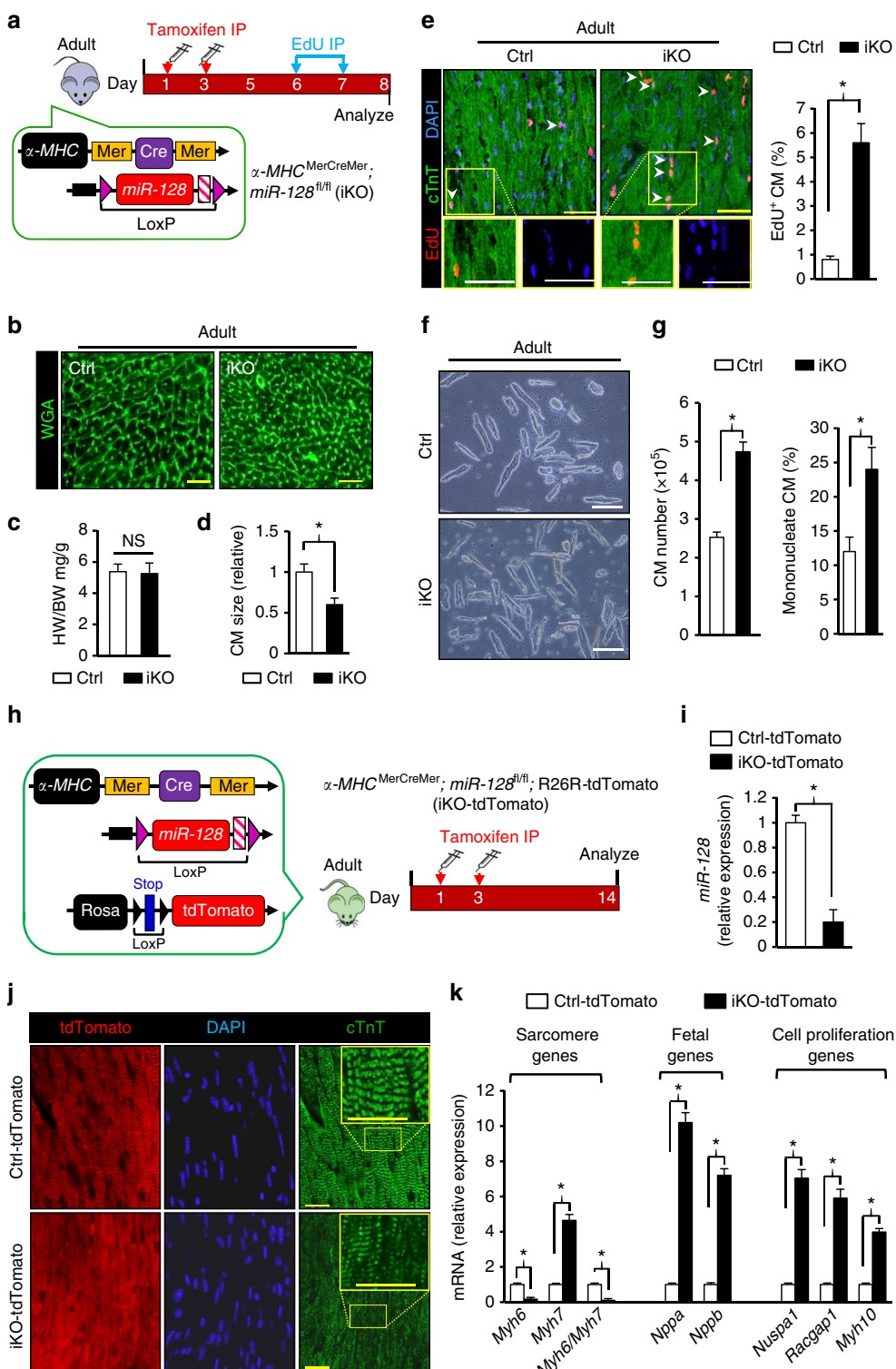

## Discussion

The application of direct activation of pre-existing CM proliferation is emerging as one of the most promising strategies in cardiac regenerative medicine[5,25,26]. Dissecting the mechanisms by which adult CMs exit the cell cycle arrest is fundamental for therapeutic manipulation to stimulate endogenous CM proliferation in the adult myocardium. Although several CM cell cycle mediators have been identified[5,25], manipulation of these genes is insufficient for full recovery of heart function in response to injury. It is, therefore, essential to discover novel therapeutic targets that activate endogenous CM proliferation and recovery of cardiac function after damage.

Involvement of miRNAs has been invoked as one mechanism underlying regulation of cell proliferation. In this report, we propose a model in which inhibition of *miR-128* in vivo promotes cardiac regeneration by activating CM proliferation. We demonstrate that (1) The upregulation of *miR-128* in heart tissue is associated with the cell cycle exit of CMs during postnatal growth. (2) Cardiac-specific overexpression of *miR-128* leads to premature cell cycle arrest and cardiac hypertrophy. (3) *miR-128* regulates several cell cycle-related genes such as *p27* by targeting *Suz12*. (4) Overexpression of *miR-128* inhibits CM proliferation and neonatal heart regeneration. (5) Inhibition of *miR-128* prolongs the postnatal CM proliferation window and enhances the cardiac regenerative capacity of adult heart.

As a neuronal-enriched miRNA[15,27], *miR-128* is associated with central nervous system development[28,29] and is downregulated in gliomas[30]. Downregulation of *miR-128* accelerates glioma-initiating neural stem cell proliferation and contributes to the development of gliomas[31]. Recent research has demonstrated the involvement of *miR-128* in cardiac repair of lower vertebrates such as the newt[32] and showed that *miR-128* inhibitors enhanced the proliferation (hyperplasia) of non-CMs and extracellular matrix deposition but had no effect on CMs, which was contrary to our current finding in mice. The discrepancy is potentially due to the use of different animal models with complex heart cell phenotypes. However, these findings bring important information into the translational study of mammals. In our study, *miR-128* was revealed for the first time to be a negative regulator of the CM cell cycle when using a cardiac lineage-restricted transgenic mouse model. Deletion of *miR-128* prolonged the postnatal CM proliferation window, as evidenced by pronounced sarcomere disassembly and proliferative markers including pH3, Ki67, Aurora B, and EdU. Normal growth of developing heart requires a proper balance between cycling cells and cells that exit the cycle. Disturbance in this balance is associated with hypertrophy[33–35] that is defined as cell enlargement due to an increase in protein or RNA content without DNA replication or cytokinesis. RNA-seq revealed that the downstream genes of *miR-128* are involved in pathways of DNA replication, cell cycle, hypertrophic cardiomyopathy, or dilated cardiomyopathy. Transgenic mice overexpressing *miR-128* displayed premature cell cycle exit, cardiac hypertrophy, and cardiac dysfunction. Further analysis is underway to explore the association of *miR-128* activation in pathogenesis of congenital heart disease involving abnormalities of myocardial growth.

Cell cycle exit in CMs is accompanied by downregulation of positive cell cycle regulators and upregulation of CDKIs[36]. Among the target genes regulated by *miR-128*, is *Suz12* whose expression is significantly lower in adult hearts. This finding suggests that *Suz12* plays a primary role in CM cell cycle regulation including neonatal cell cycle withdrawal and the later stages of heart development. This study demonstrated that knockdown of *Suz12* resulted in a reduction of CM proliferation, which is consistent with its function in gene silencing[23,37] and playing a fundamental role in mouse development[38]. Accompanied with the upregulation of SUZ12 protein, downregulation of negative cell cycle regulators (such as *p27*) and upregulation of downstream positive cell cycle regulators such as Cyclin E and CDK2 were observed in the *miR-128*-deficient heart. *P27*, as a major member of CIP/KIP CDKI family, has been implicated in CM cell cycle arrest[39] and its deletion promotes CM entry into S-phase[40–42]. *P27* can also negatively modulate the activity of *Cyclin E*-dependent kinase[36,43], a requirement for entry of cells into S-phase[44]. Our data suggest that elimination of *miR-128* might activate cell cycle-related genes, in part through SUZ12-regulated histone modification, thereby promoting CM proliferation. Since SUZ12 is a chromatin-associated protein that is broadly distributed, it is likely that it regulates other genes involved in CM proliferation. Thus, it would be informative to perform ChIP-seq and RNA-seq to systemically identify target genes of SUZ12 to better understand how SUZ12 modulates the activity of EZH2, the enrichment of H3K27me3 and transcriptional output. Recent studies on the role played by the nervous system in endogenous heart regeneration are yielding mechanistic insights. For instance, direct mechanical denervation impairs heart regeneration in mammals, but is restored upon administration of neuregulin-1 (NRG1) and nerve growth factor (NGF). Evidence from studies in zebrafish and mice has revealed that *NRG1-Erbb* signaling is crucial for proper heart formation, CM proliferation, and morphology[45–47]. Interestingly, our RNA-seq data set revealed downregulation of *Erbb2* (NRG1 co-receptor) in *miR-128*[OE] heart. However, it remains to be determined whether the effects of *miR-128* on cardiac regeneration following injury are mediated in part by its activity in the nervous system. Using a gain-of-function genetic approach in our neonatal cardiac injury model, we found that *miR-128* overexpression inhibited CM proliferation and neonatal heart regeneration. These findings highlight the involvement of *miR-128* in the pathway that arrests CM proliferation and cardiac regeneration after birth. To assess the potential therapeutic benefit of *miR-128* inhibition in MI, we generated a cardiac-specific, tamoxifen-inducible *miR-128* knockout mouse model. The effect of *miR-128* deletion on the cell cycle was evident in adult stages, when adult CMs are fully

**Fig. 7** *MiR-128* deletion promotes proliferation of adult CMs. **a** Schematic diagram depicting the protocol of tamoxifen (TAM)-inducible *miR-128* deletion (iKO) in adult hearts (P28). Control (Ctrl) mice were *miR-128*[fl/fl] mice, and iKO mice were α-MHC[MerCreMer]; *miR-128*[fl/fl] mice. **b** WGA staining in adult control and iKO hearts. Scale bars, 50 μm. **c** Measurement of HW to BW ratio in control and iKO hearts (*n* = 6). **d** Quantification of CM size determined by WGA staining (*n* = 6 mice, ~150 CMs/heart). **e** Evaluation of CM proliferation by EdU incorporation (*n* = 1130 CMs pooled from five mice). Scale bars, 25 μm. **f** Representative images of isolated adult CMs in control and iKO hearts. Scale bars, 100 μm. **g** Comparison of CM number in hearts from control and iKO mice. Approximately 2000 CMs were counted per sample, three independent samples per group. **h** Schematic diagram for the TAM-inducible dual-lineage tracing protocol for the mouse models. Control (Ctrl-tdTomato) mice were α-MHC[MerCreMer]; Rosa[tdTomato]. The iKO-tdTomato mice were αMHC[MerCreMer]; *miR-128*[fl/fl]; Rosa[tdTomato]. **i** qPCR analysis of *miR-128* expression in Ctrl-tdTomato and iKO-tdTomato hearts (*n* = 5). **j** Analysis of in vivo sarcomere structure of hears from Ctrl-tdTomato and iKO-tdTomato mice by immunofluorescence staining of cTnT (*n* = 5). Scale bars, 10 μm. **k** Expression of sarcomere genes, fetal genes, and genes associated with cell proliferation analyzed by qPCR in adult hearts from Ctrl-tdTomato and iKO-tdTomato mice (*n* = 6). Statistical significance was calculated using Student's *t*-test in **b**, **d**, **e**, **g**, **i**, and **k**. Data are represented as means ± SEM. *P < 0.05. NS, not significant

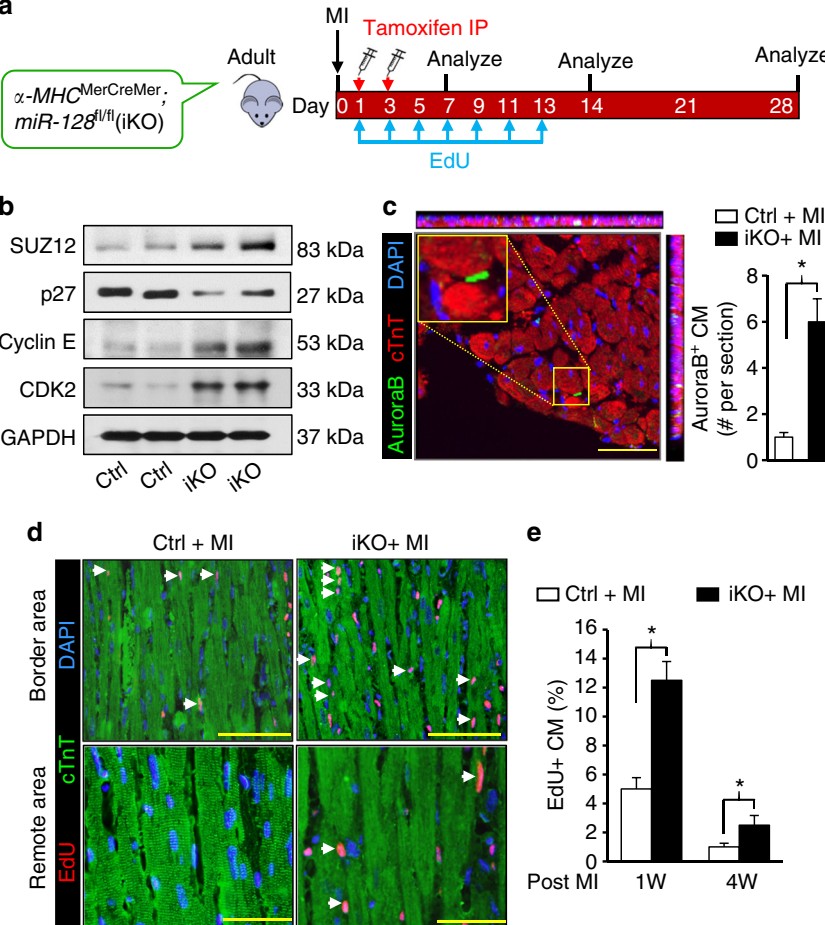

**Fig. 8** *MiR-128* deletion promotes adult cardiomyocyte proliferation after MI. **a** Schematic of the experimental design for assessing adult (12-weeks-old) cardiac regeneration following MI in TAM-inducible *miR-128* knockout (iKO) mice. Control (Ctrl) mice were *miR-128*[fl/fl] mice, and iKO mice were *α-MHC*[MerCreMer]; *miR-128*[fl/fl] mice. **b** Western blot analysis of cell cycle-related gene expression in infarcted hearts at day 7 after TAM administration (n = 3). **c** Immunostaining of Aurora B in infarcted hearts at day 14 after TAM administration (n = 5 mice, ~400 CMs/heart). Scale bars, 250 pixel. **d** Evaluation of CM proliferation by EdU incorporation assay. Scale bars, 50 μm. **e** Quantification of EdU+ CMs in the Ctrl and iKO hearts at 1 week (1 W) and 4 weeks (4 W) post MI (n = 5 mice, ~250 CMs/heart). Statistical significance was calculated using Student's *t*-test in **c**, **e**. Data are represented as means ± SEM. *$P < 0.05$

differentiated and quiescent and their ability to divide is quite limited. Using genetic lineage tracing, our results provide proof-of-concept that pre-existing CMs rather than progenitor cells are, in fact, the target cells that respond to *miR-128* deletion during regeneration after cardiac damage. These adult CMs lacking *miR-128* can be "rejuvenated" to an immature stage that allows them to dedifferentiate and enter a proliferative state, an endogenous program for natural heart regeneration that occurs in the zebrafish and neonatal mice in response to injury[48]. Of particular interest, the changes in cellular capabilities induced by loss of *miR-128* result in increased cellular plasticity that allows significant anatomical and functional capacity upon injury, but does not impair heart function under normal conditions.

In summary, our study demonstrates that we can activate endogenous CM proliferation by targeting *miR-128*, and that this strategy is a potentially valuable approach for inducing myocardial regeneration, and may lead to major therapeutic advances in the treatment of human heart disease.

## Methods

**Laboratory animals**. All research protocols conformed to the Guidelines for the Care and Use of Laboratory Animals published by the National Institutes of Health (National Academies Press, eighth edition, 2011). All animal use protocols and methods of euthanasia (pentobarbital overdose followed by thoracotomy) used in

this study were approved by the University of Cincinnati Animal Care and Use Committee. An independent review and approval of cell and chemical drug used in this study was conducted by the Institutional Biosafety Committee (IBC). Mice were maintained on a C57BL/6 background and their genotype was determined by PCR from tail DNA. Both male and female mice were randomized in different experiment groups.

**Generation of mice with conditional overexpression of *miR-128***. A construct was engineered for knockin of the *miR-128* (*miR-128-3p*) gene into the Rosa26 locus. Rosa26 genomic DNA fragments (~1.1 kb and ~4.3 kb 5′ and 3′ homology arms, respectively) were amplified from C57BL/6 BAC DNA, cloned into the pBasicLNeoL vector sequentially by in-fusion cloning, and confirmed by sequencing. The *miR-128* gene, under the control of tetO-minimum promoter, was also cloned into the vector between the two homology arms. In addition, the targeting construct also contained a loxP sites flanking the neomycin resistance gene cassette for positive selection and a diphtheria toxin A (DTA) cassette for negative selection. The construct was linearized with ClaI and electroporated into C57BL/6N ES cells. After G418 selection, seven-positive clones were identified from 121 G418-resistant clones by PCR screening. Six-positive clones were expanded and further analyzed by Southern blot analysis, among which four clones were confirmed with correct targeting with single-copy integration. Correctly targeted ES cell clones were injected into blastocysts, and the blastocysts were implanted into pseudo-pregnant mice to generate chimeras by Cyagen Biosciences Inc. Chimeric males were bred with Cre deleted mice from Jackson Laboratories to generate neomycin-free knockin mice. The correct insertion of the *miR-128* cassette and successful removal of the neomycin cassette were confirmed by PCR analysis with the primers listed in Supplementary Table 1.

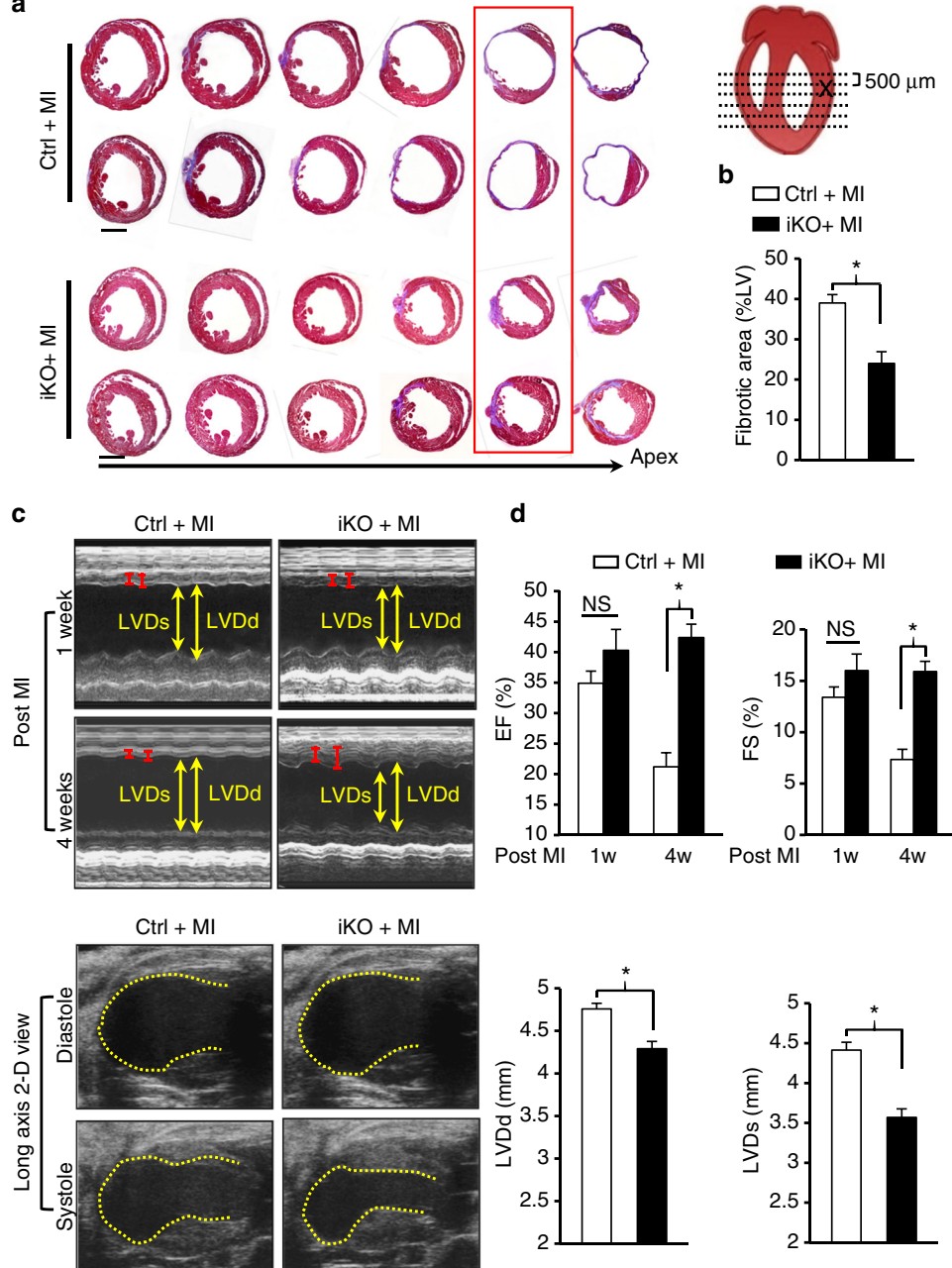

**Fig. 9** *MiR-128* deletion promotes adult cardiac regeneration after MI. **a** Representative images of Masson trichrome-stained heart section at 28 days after MI. Serial sectioning was performed at 500 μm intervals. The most significant difference between two groups is highlighted by the red box. Scale bars, 1 mm. **b** Measurement of fibrotic areas in heart sections following MI in control and iKO mice analyzed by Masson trichrome staining. (*n* = 8). **c**, **d** Heart function analyzed by echocardiography and quantified by LVDd, LVDs, EF, and FS (*n* = 6). Control (Ctrl) mice were *miR-128*$^{fl/fl}$ mice, iKO mice were *α-MHC*$^{MerCreMer}$; *miR-128*$^{fl/fl}$ mice. Statistical significance was calculated using Student's *t*-test in **b**, **d**. Data are represented as means ± SEM. *$P < 0.05$. NS, not significant

Mice with doxycycline-inducible CM-specific overexpression of *miR-128* (*miR-128-3p*) were generated by crossing *α-MHC*-tTA (The Jackson Laboratory) mice with *miR-128*$^{TetRE}$ mice, in which tetracycline-responsive transcriptional activator (tTA) expression is under the control of *α-MHC* promoter. Doxycycline (Dox, Harland Laboratories) containing diet was administered to repress transgene expression.

**Generation of mice with a conditional deletion of *miR-128*.** A construct was engineered for conditional disruption of the *miR-128* (*miR-128-3p*) gene in which a 1.7 kb fragment spanning the *miR-128* gene was flanked by two loxP sites. The 1.7 kb fragment, the 5.4 kb left homology arm, and the 2.9 kb right homology arm were amplified from C57BL/6 BAC DNA and cloned into the pBasicLFNeoFL vector sequentially by in-fusion cloning and confirmed by sequencing. In addition to

conditional knockout (cKO) region and homology arms, the targeting construct also contains Frt sites flanking the neomycin cassette for positive selection and a DTA cassette for negative selection. The construct was linearized with NotI and electroporated into C57BL/6N ES cells. After G418 selection, three-positive clones were identified from 280 G418-resistant clones by PCR screening. The positive clones were expanded and further analyzed by Southern blot. The random integration of extra copies of targeting construct was excluded by hybridization with a neomycin probe. To generate chimeras, ES cell clones were microinjected into blastocysts, and the blastocysts were implanted into pseudo-pregnant foster mice by Cyagen Biosciences Inc. Chimeric males were bred with Flp deleter mice from Jackson Laboratories to generate neomycin-free floxed mice. The correct integration of loxP sites and the successful removal of the neomycin cassette were confirmed by PCR analysis with the primer listed in Supplementary Table 1.

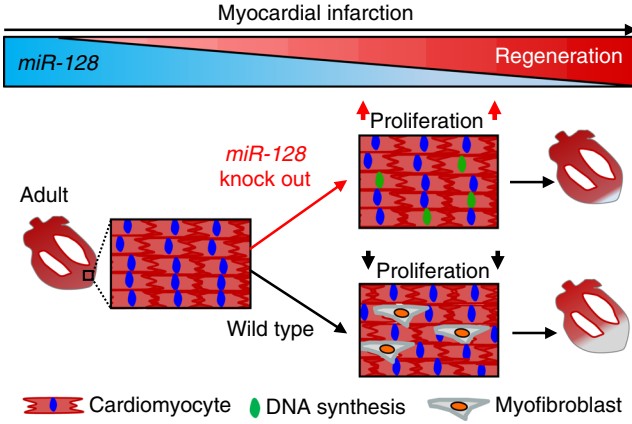

**Fig. 10** Loss of *miR-128* activates endogenous cardiac regeneration. A schematic diagram proposing that loss of *miR-128* activates cardiac regeneration by promoting cardiomyocyte proliferation, while the necrotic tissue of wild-type heart is replaced by myofibroblasts with fibrous scars in response to MI

Cardiac-specific *miR-128* knockout mice (*miR-128⁻/⁻*) were generated by crossing *Nkx2.5*^Cre (The Jackson Laboratory) mice with *miR-128*^fl/fl mice.

Tamoxifen (TAM) inducible CM-specific *miR-128* knockout mice (iKO) were generated by crossing *α-MHC*^MerCreMer mice (Tg(*α-MHC*-cre/Esr*)1Jmk/J, The Jackson Laboratory) with *miR-128*^fl/fl mice. Induction of Cre recombinase activity was achieved using two doses of tamoxifen (Sigma, 0.25 mg g⁻¹ body weight) dissolved in corn oil (Sigma) and administrated intraperitoneally (i.p.) on day 1 and 3.

A dual-lineage tracing system was established to investigate the origin of regenerated CMs. *α-MHC*^MerCreMer mice were crossed with Rosa26-tdTomato (R26R-mTmG, The Jackson Laboratory) reporter mice and *miR-128*^fl/fl mice to generate iKO-mTmG mice (*α-MHC*^MerCreMer;*miR-128*^fl/+;R26R-mTmG) to label *miR-128* null CM with red color following tamoxifen administration. All mice were maintained on a C57BL/6 background.

**RNA-seq and bioinformatics analysis**. RNA-seq experiments were performed by the Genomics, Epigenomics and Sequencing Core (GESC) at the University of Cincinnati. To analyze differential gene expression, sequence reads were aligned to the mouse genome (mm10) by using standard Illumina sequence analysis pipeline, which was performed by The Laboratory for Statistical Genomics and Systems Biology at the University of Cincinnati.

The sample processing, sequencing, and data analysis were described in previous publications[49,50]. Sample tissues were homogenized in 0.6 ml lysis/binding buffer from the *mir*Vana miRNA Isolation Kit (Thermo Fisher Scientific, AM1560) using a Bullet Blender Storm homogenizer (MIDSCI). Total RNA extract was performed according to the mirVana protocol, and the RNA was eluted with 100 µl of elution buffer. Quality of RNA was assessed using a 2100 Bioanalyzer (Agilent Technologies).

RNA-seq reads were processed to remove quality reads and then aligned to the mouse reference genome using TopHat2 aligner[51]. Reads aligning to known gene were counted using Bioconductor packages for next-generation sequencing data analysis[52]. The differential expression analysis between *miR-128*^OE hearts and control samples was performed using the negative binomial statistical model of read counts as implemented in the edgeR Bioconductor package[53]. The KEGG pathways[54] and MSigDb gene sets[55] enrichment analysis was performed using LRpath methodology as implemented in the CLEAN package[56], and ranked by the enrichment *P*-value. Functional enrichment analysis of the genes that are both downregulated in *miR-128*^OE hearts and are predicted targets of *miR-128* is performed using PANTHER enrichment analysis tool[57].

**Neonatal CM isolation and culture**. Neonatal rat CMs were isolated from ventricles of 1-day-old neonatal Sprague–Dawley rats (Harland) using a neonatal CM isolation kit (Worthington Biochemical, LK003300) according to the manufacturer's instructions. Neonatal mouse CMs were isolated from 1-day-old (P1) C57BL/6 mice with a modified protocol as previously described[58]. Briefly, after washing and mincing the neonatal mouse hearts with PBS (without Ca²⁺, Mg²⁺) supplemented with 20 mM BDM (Sigma-Aldrich), tissue fragments were incubated in the isolation medium (PBS supplemented with 20 mM BDM and 0.0125% trypsin) with gentle agitation 4 °C overnight. Predigested tissue fragments were then transferred into the freshly made digestion medium (1.5 mg ml⁻¹, collagenase/dispase mixture, Roche) and incubated for 20 min at 37 °C. The cell suspension was collected and centrifuged to yield the isolated cell pellets. Cells were plated and incubated for 2 h in cell culture incubator. The adherent fibroblast were further

cultured and harvested. The non-adherent CMs were re-suspended and cultured in 0.1% gelatin plus with 10 µg ml⁻¹ fibronectin (Sigma-Aldrich, F1141) coated slides with 68% Dulbecco's Modified Eagle's Medium (DMEM) high-glucose medium supplemented with 17% M-199, 4% horse serum (Gibco, 26050088), 10% FBS, and 1% penicillin/streptomycin (hereafter referred to as "complete-medium") at 37 °C and 5% CO2. After the cells were allowed to adhere for 24 h, *miR-128* mimic (50 nM, Dharmacon, C-310957-01-0005), *miR-128* inhibitor (50 nM, Dharmacon, IH-310957-02-0005), or siRNA against *Suz12* (50 nM, Dharmacon, L-040180-00-0020) transfection were performed according to the manufacturer's instructions. After 48 h, cells were harvested for analysis. Control samples were transfected with an equal concentration of negative control mimic, inhibitor or siRNA.

**Adult CM isolation and culture**. Adult mouse CMs were isolated from adult C57BL/6 mice with a modified protocol as described[59]. Briefly, the aorta of the excised heart was clamped and mounted on a Langendorf apparatus and perfused with calcium-free perfusion buffer (113 mM NaCl, 4.7 mM KCl, 0.6 mM KH₂PO₄, 0.6 mM Na₂HPO₄, 1.2 mM MgSO₄, 10 mM Na-HEPES, 12 mM NaHCO₃, 10 mM KHCO₃, 0.032 mM phenol red, 30 mM taurine, 10 mM BDM, and 5.5 mM glucose; pH-7.0.), followed by digestion with 50 ml perfusion buffer containing 15,000 U of type II collagenase (Roche) and 50 µM CaCl₂. Stop digestion when the heart became slightly pale and flaccid. Then the heart was gently teased into small pieces with forceps, and triturated with a Pasteur pipette to dissociate individual CM. The cell suspension was collected and centrifuged to yield CM pellets. The adult mouse CMs were cultured on laminin (10 µg ml⁻¹, Life technologies, 23017015) coated slides with AW medium (Cellutron life technologies, m-8034) with 10% FBS.

**Luciferase reporter assay**. The DNA fragment containing 3′ untranlsted regions (3′UTR) was amplified by PCR and cloned into luciferase reporter vector-psiCHECK2 (Promega, C8021). The reporter vector containing mutant 3′UTR was generated using a Site-Directed Mutagenesis Kit (New England Biolabs). HEK-293 cells (ATCC, CRL-1573) were transfected using DharmaFECT Duo reagent (Dharmacon, T-2020-01) according to the manufacturer's instructions with luciferase reporter vector and *miR-128* mimic (Dharmacon, C-310957-01-0005). Cells were harvested and assayed for luciferase activity using Dual-Glo™ kit (Promega, E2920) 48 h after transfection as previously described[60]. Cell lysates were assayed in a luminometer normalizing firefly to renilla luciferase activity. Data were expressed as percent inhibition relative to control miRNA mimic transfected cells.

**Quantitative real-time PCR (qPCR)**. Total RNA was isolated using Trizol reagent (Invitrogen), followed by DNase treatment and purification using RNeasy mini column kit (Qiagen, 74104). Complementary DNA was synthesized using miScript PCR Starter Kit (Qiagen, 218193) according to the manufacture's instruction. qPCR was performed on a CFX96 Real-Time PCR system (Bio-Rad) using the miScript PCR Starter Kit. The fold changes of each target mRNA expression relative to GAPDH under experimental and control conditions were calculated based on the threshold cycle (CT) as $r = 2 - \Delta(\Delta CT)$, where $\Delta CT = CT(target) - CT$ (GAPDH) and $\Delta(\Delta CT) = \Delta CT(experimental) - \Delta CT(control)$. The primers for qPCR are listed in Supplementary Table 1.

**Chromatin immunoprecipitation-qPCR (ChIP-qPCR) assay**. ChIP assay was performed as described previously[61]. Heart tissue was cut into small pieces and crosslinked with PBS with 1% formaldehyde (Thermo Fisher Scientific, 28906) at room temperature for 10 min. Fixation was terminated by addition of 0.125 M (final concentration) glycine (Sigma-Aldrich, 50046) at room temperature for 5 min. Chromatin was isolated by addition of lysis buffer, followed by disruption with a Dounce homogenizer. Lysates were sonicated and sheared into a size range of ~200–500 bp.

An aliquot of chromatin (30 µg) was precleared with Dynabeads Protein G (Life Technologies, 10009D) at 4 °C for 1h. The precleared chromatin was immunoprecipitated with antibodies as follows: SUZ12 (D39F6) (Cell Signaling Technology, 3737, 1:100), EZH2 (D2C9) (Cell Signaling Technology, 5246, 1:100) and H3K27me3 (C36B11) (Cell Signaling Technology, 9733, 1:50). Chromatin were decrosslinked by incubation with proteinase K at 65 °C for overnight, and ChIPed DNA was purified by MinElute kit (Qiagen, 28004). Genomic DNA (Input) was prepared from an aliquot of precleared chromatin by decrosslinking and purification. The resulting DNA was quantified on a Qubit spectrophotometer.

**Western blot analysis**. Cells were lysed with ice-cold cell lysis buffer plus protease inhibitor (Sigma-Aldrich, P8340). Protein samples (40 µg) were mixed and resolved in 4 × SDS/PAGE sample buffer and boiled for 15 min before loading onto 10% polyacrylamide gels (Bio-Rad). The electrophoresed proteins were transferred from the gel to nitrocellulose membranes (Bio-Rad). Equal loading and transfer of proteins was confirmed by Ponceau red staining. Membranes were incubated for 60 min with 5% dry milk and Tris-buffered saline to block non-specific binding sites. Membranes were immunoblotted overnight at 4 °C with antibodies against Cyclin E (M-20) (Santa Cruz, SC-481, 1:500), SUZ12 (D39F6) (Cell Signaling Technology, 3737, 1:1000), CDK2 (72B2) (Cell Signaling Technology, 2546, 1:1000), p27 (D69C12) (Cell Signaling Technology, 3686, 1:1000), PPARγ (81B8) (Cell Signaling Technology, 2443, 1:1000), and GAPDH

(Sigma-Aldrich, G9545, 1:5000) on a rocking platform. After washing three times for 5-min with Tris-buffered saline, the membranes were incubated for 60 min with HRP-conjugated secondary antibody, washed three times with Tris-buffered saline, and developed with the ECL plus kit (Thermo Scientific). Uncropped immuno-blotting images are presented in Supplementary Fig. 11.

**Neonatal mouse apex resection.** Apex resection (AR) was performed on neonatal mice on postnatal day 1 (P1) as described[62]. All the neonatal mice were anesthetized by hypothermia (on ice for 3–5 min) placing gauze below the pups. Steady pressure on the abdomen was applied to exteriorize the heart when the heart was exposed through a left thoracotomy incision. Then, the ventricular apex was resected using several incisions until the anatomical landmark of the chamber was exposed. The hearts were harvested at 6 h, 24 h, 3 days, 7 days, 14 days, and 21 days post AR. Sham-operated mouse groups (control) underwent chest opening without AR.

**Myocardial infarction (MI).** An MI model was developed in female mice, as previously described[61]. Briefly, mice (8–10 weeks old) were anesthetized by spontaneous inhalation and maintained under general anesthesia with 1–2% iso-flurane. Animals were mechanically ventilated using a rodent ventilator (Harvard Apparatus) connected to an endotracheal tube. The heart was exposed by a left side limited thoracotomy and the LAD was ligated with a 6-0 polyester suture 1 mm from the apex of the normally positioned left auricle.

**Echocardiography.** Transthoracic echocardiography (Visual Sonics Vevo 2100) was performed with a 40-MHz probe. Hearts were imaged in 2D long-axis view at the level of the greatest LV diameter in animals under light general anesthesia. This view was used to position the M-mode cursor perpendicular to the LV anterior and posterior walls. LV end-diastolic (LVDd) and end-systolic diameters (LVDs) were measured from M-mode recordings. LV EF was calculated as: EF% = [(LVDd)$^3$-(LVDs)$^3$/(LVDd)$^3 \times 100$]. LV FS was determined as: FS% = [(LVDd–LVDs)/LVDd × 100]. All measurements were performed according to the American Society for Echocardiography leading-edge technique standards, and were averaged over three consecutive cardiac cycles.

**EdU pulse-chase experiment.** For EdU (5-ethynyl-2′-deoxyuridine, Life technology) labeling experiments in vivo, animals were injected intraperitoneally (i.p.) at 200 µg g$^{-1}$ body weight. EdU staining was performed with Click-iT EdU Imaging kit (Thermo Fisher Scientific, C10337) according to the manufacturer's instructions.

**Systemic delivery of siRNA in vivo.** SiRNA (Dharmacon) against *Suz12* (si-*Suz12*, L-040180-00-0020) and siRNA control (si-Ctrl, D-001810-01-05) were formulated with MaxSuppressor™ In Vivo RNA-LANCEr II (Bioo Scientific, 3410-01) according to the manufacturer's instructions. The mice were injected i.p. at 2 µg g$^{-1}$ body weight with si-*Suz12* or si-Ctrl at P1, P3, P5, and hearts were harvested at P7.

**Analysis of left ventricular (LV) fibrotic area.** Masson's trichrome staining was performed to quantify fibrosis area in the left ventricle post injury. An Olympus BX41 microscope equipped with CCD (Magna-Fire TM) camera captured LV area images on each slide. LV fibrosis area and total LV area of each image were measured using the Image J and fibrosis area was reported as a percentage of the total LV area.

**Immunohistochemistry assay.** After deparaffinization and microwaving antigen retrieval in citric acid buffer, heart sections were incubated for 1 h at 37 °C or overnight at 4 °C with the following antibodies: Anti-cTnT antibody (13-11) (Thermo Fisher Scientific, MS-295-P1, 1:100) was used to identify CM. Anti-Ki67 (Abcam, ab15580, 1:200), anti-EdU (Life technology, C10637, 1:500), anti-phosphorylated-histone 3 (Ser10) (pH3, Millipore, 06-570, 1:100), and Aurora B (35C1) (Sigma-Aldrich, A2606, 1:100) antibodies were used to analyze cell cycle activity, DNA synthesis, karyokinesis, and cytokinesis respectively. After triple washing in PBS, slides were incubated for 45 min at 37 °C with fluorescence conjugated second antibodies (Jackson Immuno Research). For WGA staining, slides were incubated for 30 min at 37 °C with primary antibody conjugated to Alexa Fluo 488 (Thermo Fisher Scientific, W11261, 1:500) in PBS. To quantify apoptotic CMs, additional mouse hearts were subjected to TUNEL (Promega) and cTnT (Thermo Fisher Scientific, MS-295-P1, 1:100) staining according to the manufacturer's instructions. To quantify CM proliferation, cells were stained with Ki67 and pH3. DAPI was used for nuclear counterstaining. Four fields of each section were examined for quantification. Fluorescent imaging was performed with an Olympus BX41 microscope equipped with an epifluorescence attachment.

**Statistical analysis.** Results were statistically analyzed with the use of the StatView 5.0 software package (Abacus Concepts Inc., Berkeley, CA). All values are expressed as means ± SEM. Student's *t*-test was applied appropriately for comparison between two treatment groups. One-way ANOVA (using the post-hoc Bonferroni/Dunn test) was performed for comparisons of multiple groups in each of the specific experimental designs presented in the figures.

**Data availability**. All of the raw sequencing data from this study have been submitted to the NCBI GEO (Accession code: GSE107684).

All other supporting data from this study are available from the article and Supplementary Information files, or from the corresponding author upon reasonable request.

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

## Acknowledgements

This work was supported by the National Natural Science Foundation of China (Nos. 81330007, U1601227 to Dr. Xi-Yong Yu.), the National Institutes of Health grants HL107957, HL110740, HL136025, and HL130042 (to Dr. Yigang Wang), the Science and Technology Programs of Guangdong Province (No. 2014A050503047 to Dr. Xiyong Yu and Dr. Yigang Wang, 2015B020225006 to Dr. Xi-Yong Yu), Guangzhou Science and Technology Program (No. 201604010087 to Dr. Xi-Yong Yu.), and the American Heart Association Catalyst (17CCRG33671128) to Dr. Sakthivel Sadayappan and Dr. Yigang Wang. We thank Anne Schaefer (Icahn School of Medicine at Mount Sinai, USA) for providing the *miR-128*<sup>flox/flox</sup> mice and Gary E. Shull for manuscript editing.

## Author contributions

W.H. designed and performed the experiments, analyzed data, and drafted the manuscript. Y.F. designed the experiments, analyzed data, and wrote the manuscript. J.L. performed ChIP and some of the in vivo experiments, interpreted data, discussed, and edited manuscript. H.Y. participated in the histology assay. C.W. and M.W. performed some RNA-seq data analysis and PANTHER enrichment analysis. L.J., B.W., and W.C. performed the immunoblotting and some of the in vitro experiments. W.M. supplied some study materials. M.M. and J.C. performed statistical and bioinformatics analysis of RNA-seq data and assisted with interpretation of results. C.P. collected data and wrote manuscript. W.S.D., S.S., and P.J.S. performed manuscript editing. X.-Y.Y. and Y.W. (corresponding authors) designed and supervised the study, and performed manuscript editing.

## Additional information

**Competing interests:** The authors declare no competing financial interests.

