## [Peer Review File · Nature Communications]

Reviewers' comments:

Reviewer #1 (Remarks to the Author):

The manuscript by Huang and collaborators reports the role of miR-128 in cardiomyocyte proliferation during fetal life and myocardial regeneration after cardiac damage. Through the use of a remarkable number of genetically modified mice to overexpress or conditionally block this miRNA, the authors wish to prove that its major effect is to repress cardiomyocyte proliferation. Yet, the manuscript leaves the reader both unconvinced of the effect and disappointed by the major conclusions, since most of the experiments are performed superficially and lack specific controls, thus rendering the major conclusions inconsistent.

The major concerns are as follows.

The explanation why the hearts of miR-128-expressing mice in Fig. 2C and 2D are hypertrophic is unclear. Should the authors' interpretation be that miR-128 blocks cardiomyocyte proliferation and thus induces compensatory cardiomyocyte hypertrophy, why should this also result in massive organ hypertrophy?

Fig. 1G-H. The values for EF and FS are remarkably high. Do normal hearts in the authors' experimental condition have an EF of 90%? The authors should show other echocardiography parameters of cardiac function to sustain this finding.

The authors state that (p. 6): "Interestingly, miR-128OE mice displayed pathologically dilated cardiomyopathy that was consistent with focal replacement fibrosis, CM hypertrophy, and severe heart failure compared with Ctrl mice at the same adult stage (data not published)." This information is not consistent with what shown in Fig. 2. Given the relevance of these findings, these should be presented in detail in the manuscript.

The author state that, upon treatment with anti-miR-128, cardiomyocyte became "dedifferentiated" based on immunostaining for cardiac troponin T. It is unclear how the authors can define a "differentiated" or "undifferentiated" state based on the images shown in Suppl. Fig. 3. What are the cells with elongated filaments they define as "dedifferentiated"? What are the majority of cells in these cultures, which do not stain with the anti-TnT antibody?

Suppl. Fig. 3E and 3G. Staining for phospho-H3 is not convincing. The number of positive cells is astonishingly high (more than 15%) for a marker that identifies cells that only transiently travel through the G2M phases of the cell cycle. The pictures show high magnification of a single cell, which is totally anecdotal and not significant.

Fig. 3G-H: For a heart with such a marked hyperplasty as that shown in Fig. 3C, one would expect a remarkably high proliferation activity, while it appears that the number of cycling cells is only 2fold than that of controls. This is in sharp disagreement with the picture shown in Fig. 3G (by the way, in this picture the actual cell reactivity to Ki67 is covered by the arrow marks). In the same picture the representative image is rather ambiguous. Is this a cardiomyocyte nucleus? What is the red halo around the nucleus?? Again, it is unclear how quantification of "disassembled" cardiomyocytes was performed, and certainly this value should not be expressed as "# per section" - out of how many analyzed cells?

The authors wish to identify targets for miR-128 action and reach the conclusion that Suz12 as one of these. Why this one only? There are a number of already described targets for this miRNA and many more are predicted. This information needs to be reported and a more systematic analyses has to be conducted. It is hard to believe that the authors by chance picked up the only gene that mediates the effect of the miRNA based on notoriously fallacious prediction analysis!

Figure 6F. Quantification of the number of cardiac myocytes cannot be based on the assessment of those that are isolated, especially when the difference is so relatively small between KO and controls

Other minor issues

Fig 1 panels A-C show the normal heart development at birth. This is textbook information, there is no need to show it

A previous report indicates that miR-128 is elevated in both cardiomyocytes and non-cardiomyocytes close to the regenerating zone during newt cardiac regeneration (Witman et al. Dev Biol 2013). Since apical regeneration in mice is believed to mimic fish and amphibian heart regeneration, this information appears relevant and needs to be compared with the findings reported in the manuscript.

Fig. 2B. The graph likely shows the levels of miR-128 with or without Dox induction. This has to be indicated in the legend.

Reviewer #2 (Remarks to the Author):

This study reports that miR-128 is induced in the postnatal heart and diminishes the regenerative activity of the heart. These key findings are based on loss and gain of function studies in vivo demonstrating an important effect of this miRNA on cardiomyocyte dedifferentiation/proliferation. Using adult mice, the deletion of miR-128 in cardiomyocytes increased the regenerative window and improved healing after myocardial infarctions. The authors additionally provide evidence that miR-128 is targeting SUZ12, and that this regulates the expression of the cell cycle inhibitor p27. The study is novel and well performed. I only have a few specific comments.

Specific concerns

1. Figure 2H: the EF appears very high (90%) and is reduced to about 80 %, which would be normal in adult mice. Are these measurements correct or related to the measurement at P1? Are there any confounding effects on anesthesia/technical limitation of echo at these very early stages?
2. Figure 3F: please provide cell numbers for CM (a reduction of size does not necessarily indicate proliferation).
3. Cardiomyocyte proliferation should be documented in the therapeutic study by using Aurora Kinase staining.
4. The authors only investigated and named SUZ12 as target for miR-128. It is well known that miRNAs do not work by inhibiting one target. Although this reviewer understands that it is beyond the scope of the present study to address the full mechanism, the data showing regulation of miR-128 target should be provided as suppl. Figure. It would be essential to know how many miR-128 targets are regulated and how these might influence cell cycle progression/dedifferentiation.
5. miR-128 is highly expressed in neurons. Since innervation was shown to drive cardiac regeneration, the authors might wish to speculate about a putative function of miR-128 in this process.
7. The authors failed to cite and discuss previous publication on miR-128 effects in the heart: Witman N, Heigwer J, Thaler B, Lui WO, Morrison JI. miR-128 regulates non-myocyte hyperplasia, deposition of extracellular matrix and Islet1 expression during newt cardiac regeneration. Dev Biol. 2013 Nov 15;383(2):253-63.
Zeng XC, Li L, Wen H, Bi Q. MicroRNA-128 inhibition attenuates myocardial ischemia/reperfusion injury-induced cardiomyocyte apoptosis by the targeted activation of peroxisome proliferator-activated receptor gamma. Mol Med Rep. 2016 Jul;14(1):129-36.

Reviewer #3 (Remarks to the Author):

The manuscript by Huang et al., describes their work to show how miR-128 suppresses cardiomyocyte cell cycle activity in cardiac homeostasis and disease. miR-128 is expressed at low levels in neonatal hearts (P1) and elevated at P7 and P28. In a mouse model with cardiomyocyte(CM)-specific overexpression of miR-128, they find that heart size is enlarged at P1 with increased CM size and CM proliferation. Deletion of miR-128 from Nkx2-5Cre lineage during cardiogenesis did not affect heart function at P7, although CM size was decreased and CM cell cycle activity was increased at P7 and P14. Using a bioinformatics approach, authors search for predicted target genes underexpressed in adult heart relative to neonatal heart and identify Suz12 as a potential direct miR-128 target. The direct regulation of Suz12 mRNA by miR-128 is shown in in vitro assays, and knockdown of Suz12 in Nkx-25Cre;miR-128 mutants rescues the proliferation phenotype. The authors then go on to test if miR-128 may interfere with cardiac regeneration after injury, using the neonatal apical resection model.

In cardiomyocyte miR-128 overexpression mice, neonatal hearts do not regenerate to the same extent as in control mice, and cardiomyocyte proliferation is decreased. Furthermore, it is shown that after myocardial infarction in adult mice, loss of miR-128 enhances cardiac function and suppresses tissue fibrosis.

Comments

- Previous work has shown a direct regulation of Suz12 by miR-128 (Peruzzi et al., Neuro Oncol 2013), it is suggested to cite this paper. Moreover, work from the Pu lab has demonstrated the relevance of the PRC2 complex for cardiomyocyte development (He et al., Circ Res 2012). Since this study also uses nkx2-5Cre to interfere with PRC2 complex function, it is recommended authors discuss this paper.
- For neonatal hearts and isolated CM, the authors propose that miR-128 suppresses Suz12 leading to decreased p27, cyclinE and Cdk1 in neonatal hearts and isolated CM. However, to evaluate cell cycle activity in miR-128 iKO heart after MI, the authors decide to use a different panel of genes compared to previous experiments in the manuscript. Can the authors show p27, CyclinE and Cdk2 levels in mutants versus controls? What happens to Suz12 levels in this experiment?
- What is the baseline expression level of miR-128 in the heart and which cell types is it expressed in?
- The authors claim that miR-128 regulates neonatal cell cycle withdrawal. However, for their overexpression and loss of function models, miR-128 levels are manipulated from embryonic stages onwards. Is miR-128 also expressed during development, and can the authors show at which timepoint during development miR-128 levels are changed in their overexpression and loss of function models? An alternative explanation to the observation that CM size is reduced and cell cycle activity is enhanced in the loss of function model is that embryonic CM do not mature in the absence of miR-128 and thus retain an embryonic or fetal proliferative phenotype. Previous studies using nkx2-5Cre to ablate the PRC2 complex subunit Ezh2 indicate requirements of this complex for CM development. As such the current study does not formally prove that miR-128 in heart directly regulates neonatal cell cycle withdrawal, as this observation may be secondary to embryonic onset defects. Can the authors comment on this?
- Authors report that miR-128 mice display dilated cardiomyopathy consistent with focal replacement fibrosis, CM hypertrophy and severe heart failure in adults. Can data be provided to substantiate these claims?
- EdU incorporation is quantified by counting the number of EdU+ CM per section. Because the authors also report a change in cell size between conditions, these quantifications should be corrected for the total number of CM per section.
- Authors use a datamining approach to predict miR-128 target genes that are underexpressed in adult heart versus neonatal heart. Which dataset was used to establish differential expression?
- Resolution of immuno images is not great - for instance for figure 1c, Figure 2i, Figure 6b: it can

not be appreciated whether nuclei are CM nuclei or adjacent fibroblasts; for 6b: are we to believe that nearly all EdU+ cells are CM?

– The exact genotype and treatment of controls is often missing, making it impossible to evaluate if the right controls were used. Please specify the genotype, and treatment (for instance with/without tamoxifen) of control and mutant for each experiment, for instance in figure legends.

– The rescue experiment using Siz12 siRNA in miR-128 mutant hearts is not well described, please provide details.

– The authors should discuss the recent paper by Zeng et al. showing that miR-128 inhibition during IR induced injury blocks cardiomyocyte apoptosis through the regulation of PPARG. What happens to apoptosis and PPARG in the studies presented in the current manuscript?

To Reviewer #1: We appreciate the reviewer's comments, which were very helpful in improving the manuscript.

The manuscript by Huang and collaborators reports the role of *miR-128* in cardiomyocyte proliferation during fetal life and myocardial regeneration after cardiac damage. Through the use of a remarkable number of genetically modified mice to overexpress or conditionally block this miRNA, the authors wish to prove that its major effect is to repress cardiomyocyte proliferation. Yet, the manuscript leaves the reader both unconvinced of the effect and disappointed by the major conclusions, since most of the experiments are performed superficially and lack specific controls, thus rendering the major conclusions inconsistent.

The major concerns are as follows.

Reviewer comment 1: The explanation why the hearts of *miR-128*-expressing mice in Fig. 2C and 2D are hypertrophic is unclear. Should the authors' interpretation be that *miR-128* blocks cardiomyocyte proliferation and thus induces compensatory cardiomyocyte hypertrophy, why should this also result in massive organ hypertrophy?

Author's Reply: Transgenic mice with cardiomyocyte (CM) specific *miR-128* overexpression (*miR-128*^{OE}) displayed cardiac hypertrophy and cardiac dysfunction, which coincides with a suppressed CM cell cycle. Our RNA-seq data supported this, revealing that the differentially expressed genes in *miR-128*^{OE} hearts belong to pathways involved in DNA replication, cell cycle, hypertrophic cardiomyopathy, and dilated cardiomyopathy (**Supplementary Fig. 3**). Therefore, we proposed that the suppressed CM cell cycle induced by *miR-128* overexpression might be associated with the compensatory CM hypertrophy. Although the mechanism of cardiac hypertrophy is not fully understood, our ongoing studies include investigation into the mechanisms involved in *miR-128* induced cardiac hypertrophy.

The **discussion** section has been modified accordingly on page 17-18:

'Cell hypertrophy is defined as cell enlargement due to an increase in protein and RNA content without DNA replication. KEGG pathway analysis revealed that the differentially expressed genes in *miR-128*^{OE} hearts belong to pathways involved in DNA replication, cell cycle, hypertrophic cardiomyopathy, and dilated cardiomyopathy. Transgenic mice overexpressing *miR-128* displayed premature cell cycle exit, cardiac hypertrophy, and cardiac dysfunction. Further analysis is underway to explore the association of *miR-128* activation in pathogenesis of congenital heart disease involving abnormalities of myocardial growth.'

Reviewer comment 2: Fig. 2G-H. The values for EF and FS are remarkably high. Do normal hearts in the authors' experimental condition have an EF of 90%? The authors should show other echocardiography parameters of cardiac function to sustain this finding.

Author's Reply: The values for EF (~89%) and FS (~53%) in **Figs. 2G-H** are based on a neonatal (postnatal day 1, P1) mouse model that are consistent with recent reports (*Nat Genet. 2015; 47(7): 776-83. J Vis Exp. 2016; 111*). To analyze heart function by echocardiography, mouse pups (P1) were restrained by hand in the absence of anesthesia drugs. Gentle pressure was then used to apply an ultrasound probe to the pup's chest, eliminating the common anesthesia/technical limitations. To sustain this finding, we have added additional echocardiography parameters (left ventricular diastolic diameter-LVDd and left ventricular systolic diameter-LVDs) in **Figs. 2G and H**. EF was calculated as: $EF\% = [(LVDd)^3 - (LVDs)^3] / (LVDd)^3 \times 100$. FS was determined as: $FS\% = [(LVDd - LVDs) / LVDd] \times 100$.

Reviewer comment 3: The authors state that (p. 6): "Interestingly, *miR-128^{OE}* mice displayed pathologically dilated cardiomyopathy that was consistent with focal replacement fibrosis, CM hypertrophy, and severe heart failure compared with Ctrl mice at the same adult stage (data not published)." This information is not consistent with what shown in Fig. 2. Given the relevance of these findings, these should be presented in detail in the manuscript.

Author's Reply: As suggested, additional details of our phenotype observations in adult *miR-128^{OE}* mice have been provided (**Supplementary Figs. 2D-F, Supplementary Fig. 3**).

Accordingly, this statement has been modified in the **results** section on page 7:

'To study the role of *miR-128* in heart development, *miR-128^{OE}* mice were mated in the absence of Dox (**Supplementary Fig. 2D**). Assessment of *miR-128* level by qPCR confirmed its marked overexpression by embryonic day 10.5 (E10.5) in the hearts of *miR-128^{OE}* mice (**Supplementary Fig. 2E**). These *miR-128^{OE}* mutant mice displayed enlarged heart chambers, myocardial fibrosis, CM hypertrophy, and impaired left ventricular systolic heart function at P28 (**Supplementary Figs. 2F-H**). Moreover, KEGG pathway analysis showed that oxidative phosphorylation, metabolism, hypertrophic cardiomyopathy, and dilated cardiomyopathy pathways were enriched in *miR-128^{OE}* hearts. Concomitantly, cell cycle and DNA replication pathways were suppressed in *miR-128^{OE}* hearts (**Supplementary Fig. 3**). Taken together, these data indicate that CM-specific overexpression of *miR-128* induces early CM cell cycle exit, compensatory pathological growth of CM (hypertrophy), and impaired cardiac homeostasis.'

Reviewer comment 4: The author state that, upon treatment with anti-*miR-128*, cardiomyocyte became dedifferentiated" based on immunostaining for cardiac troponin T. It is unclear how the authors can define a "differentiated" or "undifferentiated" state based on the images shown in Suppl. Fig. 3. What are the cells with elongated filaments they define as "dedifferentiated"?? What are the majority of cells in these cultures, which do not stain with the anti-TnT antibody?

Author's Reply: We appreciate this comment. Accordingly, new images with higher resolution were presented in **Supplementary Fig. 4** (previously **Supplementary Fig. 3**). Dedifferentiation can be characterized by partially disorganized sarcomere structure with cell proliferation markers and progenitor markers (Ref. 4, 8). Our results showed that inhibition of *miR-128* resulted in dedifferentiation of CM (including disassembly and reduction of sarcomere structure) (**Supplementary Fig. 4C**), and increased expression of proliferation marker such as pH3, Aurora B, and EdU (**Supplementary Figs. 4F-H**). GATA4 recently was used to identify the dedifferentiated CM (Ref. 8). In **Supplementary Fig. 4I**, an increased amount of GATA4 expression was observed in the anti-*miR-128* treated CM. Some cTnT-negative cells were also observed in **Supplementary Fig. 4** due to the limitation of CM isolation techniques. The majority of these cells were mesenchymal cells (such as fibroblast cells). Despite of the presence of cTnT-negative cells, we only focus on the dedifferentiated cTnT⁺ CMs, which were characterized by sarcomere disassembly.

Reviewer comment 5: Suppl. Fig. 3E and 3G. Staining for phospho-H3 is not convincing. The number of positive cells is astonishingly high (more than 15%) for a marker that identifies cells that only transiently travel through the G2M phases of the cell cycle. The pictures show high magnification of a single cell, which is totally anecdotal and not significant.

Author's Reply: We appreciate this comment. New data has been presented in **Supplementary Fig. 4** (previously **Supplementary Fig. 3**). To investigate the effect of anti-*miR-128* treatment on CM proliferation, analysis methods included phospho-H3 (pH3) staining (**Supplementary Fig. 4F**), Aurora B staining (**Supplementary Fig. 4G**), and EdU incorporation assay (**Supplementary Fig. 4H**). In addition, the number of EdU, pH3, and Aurora B positive CM was re-analyzed with respect to percentage of cells rather than the number per section.

Reviewer comment 6: Fig. 3G-H: For a heart with such a marked hyperplastic as that shown in Fig. 3C, one would expect a remarkably high proliferation activity, while it appears that the number of cycling cells is only 2 fold than that of controls. This is in sharp disagreement with the picture shown in Fig. 3G (by the way, in this picture the actual cell reactivity to Ki67 is covered by the arrow marks). In the same picture the representative image is rather ambiguous. Is this a cardiomyocyte nucleus? What is the red halo around the nucleus?? Again, it is unclear how quantification of "disassembled" cardiomyocytes was performed, and certainly this value should not be expressed as "# per section"- out of how many analyzed cells?

Author's Reply: In the previous version of **Fig. 3G**, yellow arrow marks were used to indicate the sarcomere disassembled CMs rather than the cell proliferation activity. As suggested, we avoided confusion by presenting new higher resolution images in **Fig. 3G**. Yellow arrows in the new images are indicative of Ki67⁺ CM with sarcomere disassembly. In **Figs. 3G-5 and G-6**, the CM dedifferentiation phenotype was characterized by disorganized sarcomere structure as identified by cTnT immunofluorescence staining (red color around the nucleus). As suggested,

all quantification data is now expressed as percentage (%) of cells. Finally, the number of CMs we analyzed was added to the figure legend accordingly.

Reviewer comment 7: The authors wish to identify targets for *miR-128* action and reach the conclusion that *Suz12* as one of these. Why this one only? There are a number of already described targets for this miRNA and many more are predicted. This information needs to be reported and a more systematic analyses has to be conducted. It is hard to believe that the authors by chance picked up the only gene that mediates the effect of the miRNA based on notoriously fallacious prediction analysis!

Author's Reply: As suggested, we have added systematic analyses of the *miR-128* target gene screening (**Supplementary Fig. 7**).

Accordingly, this statement has been modified in the **results** section on pages 9-10:

‘RNA-Seq was performed on control (Ctrl) and *miR-128*^{OE} hearts (P7) to identify the putative target genes of *miR-128* responsible for cell cycle regulation. By comparing the downregulated mRNAs identified in *miR-128*^{OE} hearts relative to Ctrl hearts with all possible predicted candidate *miR-128* target genes¹⁹, we found 87 genes that contained the predicted binding site at the 3’UTR (**Supplementary Fig. 7A**). Gene Ontology (GO) PANTHER Analysis was then performed to identify the affected cellular biological processes. The leading biological category was ‘cellular process’ category, with nearly 28.7% of all associated genes (GO: 0009987, **Supplementary Fig. 7B**). A further subgroup analysis of the ‘cellular process’ indicated the potential for *miR-128* to affect multiple pathways that are related to regulation of the cell cycle, cell communication, and cellular component movement (**Supplementary Fig. 7B**). Moreover, the analysis of genes downregulated in *miR-128*^{OE} showed statistically significant enrichment of genes downregulated after siRNA inhibition of components of polycomb repressive complex 2 (PRC2), *Suz12* in particular (**Supplementary Fig. 7C**). PRC2 is a major epigenetic modifier that affects multiple genes and is crucial for organogenesis. Perturbation of the epigenetic landscape during early cardiac development inhibits CM proliferation, and eventually leads to fatal cardiac malformations^{20, 21}. Significantly, *Suz12* was identified among the predicted downstream target genes of *miR-128*. In contrast to neonatal hearts, the protein levels of SUZ12 was lower in the adult heart (where the CM proliferation ability is quite limited) (**Figs. 4B-C**), paralleling the upregulation of *miR-128*. These data suggested a potential interaction between *miR-128* and *Suz12*, and was a key factor in generating our hypothesis that *miR-128* regulates CM proliferation in part through the PRC2-*Suz12* signaling pathway.’

Reviewer comment 8: Figure 6F. Quantification of the number of cardiac myocytes cannot be based on the assessment of those that are isolated, especially when the difference is so relatively small between KO and controls

Author's Reply: Currently, there is no ideal approach to precisely quantify the CM number in the whole heart. As described in recent reports (*Cell*. 2014 May 8; 157(4): 795–807), isolated CMs were counted with a hemocytometer to determine the total CM number. Samples (10 μ l; 80-160 CMs/aliquot) were loaded in the counting chamber with a wide-bore pipette and counted using a hemocytometer (3 different counts/sample and 3 hearts/group). **Fig. 7F-G** (previously **Fig. 6F-G**) demonstrates that the number of adult CMs isolated in each preparation were significantly increased in iKO hearts as compared with Ctrl hearts ($\sim 2.5 \times 10^5$ for Ctrl, and $\sim 4.7 \times 10^5$ for iKO). Importantly, CM size was smaller in iKO hearts, but the heart weight to body weight ratio (HB/WB) of the iKO mice was not changed as compared to Ctrl mice (**Fig. 7C**, previously **Fig. 6C**). These data show significant differences between iKO and Ctrl mice. In addition, **Fig. 7G** (previously **Fig. 6G**) shows a significant increase in mono-nucleated CMs in iKO hearts compared with Ctrl ($\sim 12\%$ for Ctrl, and $\sim 24\%$ for iKO). These mononucleated CMs retain proliferative ability and are capable of cardiac regeneration (*Nature*. 2013; 493:433–436).

Reviewer comment 9: Fig 1 panels A-C show the normal heart development at birth. This is textbook information, there is no need to show it

Author's Reply: As suggested by Reviewer 1 and Reviewer 3, **Fig. 1A** has been deleted and new images are provided in new **Fig. 1A**.

Reviewer comment 10: A previous report indicates that *miR-128* is elevated in both cardiomyocytes and non-cardiomyocytes close to the regenerating zone during newt cardiac regeneration (Witman et al. *Dev Biol* 2013). Since apical regeneration in mice is believed to mimic fish and amphibian heart regeneration, this information appears relevant and needs to be compared with the findings reported in the manuscript.

Author's Reply: We appreciate this comment on the previous finding of *miR-128* in heart regeneration. Witman et al. demonstrated the involvement of *miR-128* in a newt model of heart injury. Their loss-of-function experiments showed that *miR-128* inhibitors can enhance the proliferation (hyperplasia) of non-myocytes and extracellular matrix deposition. Unexpectedly, they found that *miR-128* inhibitors had no significant effect on CMs in newt, which is contrary to our finding in mouse. The discrepancy is likely due to the use of different animal models with complex heart cell phenotypes. However, these findings bring important information into the translational study of mammals. In the present study, the expression of *miR-128* was modified using cardiac lineage-restricted transgene models which can exclude the potential effect of *miR-128* on non-myocytes. In addition to the apical surgery model, LAD ligation was further performed to confirm the therapeutic effect of *miR-128* inhibitor on MI mice. These findings will provide valuable insights into the design and development of targeted *miR-128* approaches for the purpose of safety and specificity in gene therapy.

Accordingly, the following comment has been added in the **discussion** section (page 17):

‘Recently published research has demonstrated the involvement of *miR-128* in cardiac repair in lower vertebrates such as the newt³¹. Their loss-of-function experiments showed that *miR-128* inhibitors enhanced the proliferation (hyperplasia) of non-myocytes and extracellular matrix deposition. Unexpectedly, they found that *miR-128* inhibitors had no significant effect on CMs in newt, which is contrary to our finding in mouse. The discrepancy is potentially due in part to the use of different animal models with complex heart cell phenotypes. However, these findings bring important information into the translational study of mammals. In the present study, *miR-128* was revealed for the first time to be a negative regulator of the CM cell cycle when using a cardiac lineage-restricted transgenic mouse model.’

Reviewer comment 11: Fig. 2b. The graph likely shows the levels of *miR-128* with or without Dox induction. This has to be indicated in the legend.

Author's Reply: The description of *miR-128* expression level in Ctrl (*miR-128*^{TetRE} mice) and *miR-128*^{OE} (α -MHC-tTA; *miR-128*^{TetRE}) hearts is indicated in the legend of **Fig. 2B** as stated in the following:

‘Fig. 2B Experimental design for CM-specific overexpression of *miR-128* at P1 (left). At right is the qPCR analysis of *miR-128* expression in control *miR-128*^{TetRE} mice (Ctrl) and *miR-128* overexpressing mice (*miR-128*^{OE}).’

To Reviewer #2: We appreciate the reviewer’s comments, which were very helpful in improving the manuscript.

This study reports that *miR-128* is induced in the postnatal heart and diminishes the regenerative activity of the heart. These key findings are based on loss and gain of function studies in vivo demonstrating an important effect of this miRNA on cardiomyocyte dedifferentiation/proliferation. Using adult mice, the deletion of *miR-128* in cardiomyocytes increased the regenerative window and improved healing after myocardial infarctions. The authors additionally provide evidence that *miR-128* is targeting SUZ12, and that this regulates the expression of the cell cycle inhibitor p27. The study is novel and well performed. I only have a few specific comments.

Reviewer comment 1: Figure 2H: the EF appears very high (90%) and is reduced to about 80 %, which would be normal in adult mice. Are these measurements correct or related to the measurement at P1? Are there are confounding effects on anesthesia/technical limitation of echo at these very early stages?

Author's Reply: The values for EF and FS in **Fig. 2H** are based on a neonatal (postnatal day 1) mouse model. To analyze heart function by echocardiography, mouse pups (P1) were

restrained by hand in the absence of anesthesia drugs, and gentle pressure was used to apply the ultrasound probe to the pup's chest. As suggested, we have included additional echocardiography parameters (left ventricular diastolic diameter-LVdD and left ventricular systolic diameter-LVDs) of cardiac function in the **Figs. 2G and H**.

Reviewer comment 2: Figure 3F: please provide cell numbers for CM (a reduction of size does not necessarily indicate proliferation).

Author's Reply: The heart weight-to-body weight ratio (HB/WB) of *miR-128*^{-/-} (*Nkx2.5*^{Cre}; *miR-128*^{fl/fl}) and Ctrl (*miR-128*^{fl/fl}) mice at P7 was statistically the same (**Fig. 3D**), but CMs in *miR-128*^{-/-} hearts were smaller (**Fig. 3F**). This could indicate an increased number of CMs in *miR-128*^{-/-} hearts. To test this proposition, CM proliferation was further analyzed by Ki67 staining and EdU incorporation assay. The results in **Fig. 3G-N** further demonstrated that knockdown of *miR-128* resulted in a significant increase in CM proliferation as evidenced by significantly increased Ki67⁺CM and EdU⁺CM. To better show the reduction of CM size in *miR-128*^{-/-} hearts, we presented new immunofluorescence images of wheat germ agglutinin (WGA) and cTnT staining of hearts at P7 in **Fig. 3F**.

Reviewer comment 3: cardiomyocyte proliferation should be documented in the therapeutic study by using Aurora Kinase staining.

Author's Reply: As suggested, CM proliferation in the adult heart post myocardial infarction (MI) was analyzed by Aurora Kinase B staining. Accordingly, the new results have been added as shown in **Fig. 8C**.

Reviewer comment 4: The authors only investigated and named SUZ12 as target for *miR-128*. It is well known that miRNAs do not work by inhibiting one target. Although this reviewer understands that it is beyond the scope of the present study to address the full mechanism, the data showing regulation of *miR-128* target should be provided as suppl. Figure. It would be essential to know how many *miR-128* targets are regulated and how these might influence cell cycle progression/dedifferentiation.

Author's Reply: Next generation RNA-Seq was conducted on *miR-128*^{OE} and control hearts (**Supplementary Fig. 7**) and as suggested, we have added systematic analyses of the *miR-128* target gene screening.

Accordingly, this statement has been modified in the **results** section on pages 9-10:

'RNA-Seq was performed on control (Ctrl) and *miR-128*^{OE} hearts (P7) to identify the putative target genes of *miR-128* responsible for cell cycle regulation. By comparing the downregulated mRNAs identified in *miR-128*^{OE} hearts relative to Ctrl hearts with all possible predicted candidate *miR-128* target genes¹⁹, we found 87 genes that contained the predicted binding site at the 3'UTR (**Supplementary Fig. 7A**). Gene Ontology (GO) PANTHER

Analysis was then performed to identify the affected cellular biological processes. The leading biological category was 'cellular process' category, with nearly 28.7% of all associated genes (GO: 0009987, **Supplementary Fig. 7B**). A further subgroup analysis of the 'cellular process' indicated the potential for *miR-128* to affect multiple pathways that are related to regulation of the cell cycle, cell communication, and cellular component movement (**Supplementary Fig. 7B**). Moreover, the analysis of genes downregulated in *miR-128*^{OE} showed statistically significant enrichment of genes downregulated after siRNA inhibition of components of polycomb repressive complex 2 (PRC2), *Suz12* in particular (**Supplementary Fig. 7C**). PRC2 is a major epigenetic modifier that affects multiple genes and is crucial for organogenesis. Perturbation of the epigenetic landscape during early cardiac development inhibits CM proliferation, and eventually leads to fatal cardiac malformations^{20, 21}. Significantly, *Suz12* was identified among the predicted downstream target genes of *miR-128*. In contrast to neonatal hearts, the protein levels of SUZ12 was lower in the adult heart (where the CM proliferation ability is quite limited) (**Figs. 4B-C**), paralleling the upregulation of *miR-128*. These data suggested a potential interaction between *miR-128* and *Suz12*, and was a key factor in generating our hypothesis that *miR-128* regulates CM proliferation in part through the PRC2-*Suz12* signaling pathway.'

Reviewer comment 5: *miR-128* is highly expressed in neurons. Since innervation was shown to drive cardiac regeneration, the authors might wish to speculate about a putative function of *miR-128* in this process.

Author's Reply: As suggested, the following comment has been added in the **discussion** section (page 17):

'Recently published research has demonstrated the involvement of *miR-128* in cardiac repair in lower vertebrates such as the newt³¹. Their loss-of-function experiments showed that *miR-128* inhibitors enhanced the proliferation (hyperplasia) of non-myocytes and extracellular matrix deposition. Unexpectedly, they found that *miR-128* inhibitors had no significant effect on CMs in newt, which is contrary to our finding in mouse. The discrepancy is potentially due in part to the use of different animal models with complex heart cell phenotypes. However, these findings bring important information into the translational study of mammals. In the present study, *miR-128* was revealed for the first time to be a negative regulator of the CM cell cycle when using a cardiac lineage-restricted transgenic mouse model.'

Reviewer comment 6: The authors failed to cite and discuss previous publication on *miR-128* effects in the heart:

Witman N, Heigwer J, Thaler B, Lui WO, Morrison JL. miR-128 regulates non-myocyte hyperplasia, deposition of extracellular matrix and Islet I expression during newt cardiac regeneration.

Dev Biol. 2013 Nov 15; 383(2):253-63.

Zeng XC, Li L, Wen H, Bi Q. MicroRNA-128 inhibition attenuates myocardial ischemia/reperfusion injury-induced cardiomyocyte apoptosis by the targeted activation of peroxisome proliferator-activated receptor gamma. Mol Med Rep. 2016 Jul; 14(1):129-36.

Author's Reply: As suggested, the comment regarding previous publication by Witman N et al., was addressed with the following statement (page 17):

'Recently published research has demonstrated the involvement of *miR-128* in cardiac repair in lower vertebrates such as the newt³¹. Their loss-of-function experiments showed that *miR-128* inhibitors enhanced the proliferation (hyperplasia) of non-myocytes and extracellular matrix deposition. Unexpectedly, they found that *miR-128* inhibitors had no significant effect on CMs in newt, which is contrary to our finding in mouse. The discrepancy is potentially due in part to the use of different animal models with complex heart cell phenotypes. However, these findings bring important information into the translational study of mammals. In the present study, *miR-128* was revealed for the first time to be a negative regulator of the CM cell cycle when using a cardiac lineage-restricted transgenic mouse model.'

The comment regarding the publication by Zeng XC et al., was added to page 15 as follows:

'Although it was previously reported that *miR-128* regulates apoptosis by targeting peroxisome proliferator-activated receptor gamma (*PPARγ*)²³, we found no significant differences in either *PPARγ* expression or apoptosis in iKO hearts when compared to Ctrl hearts at day 7 after TAM injection (**Supplementary Figs. 10C and D**).'

To Reviewer #3: We appreciate the reviewer's comments, which were very helpful in improving the manuscript.

The manuscript by Huang et al., describes their work to show how *miR-128* suppresses cardiomyocyte cell cycle activity in cardiac homeostasis and disease. *MiR-128* is expressed at low levels in neonatal hearts (P1) and elevated at P7 and P28. In a mouse model with cardiomyocyte (CM)-specific overexpression of *miR-128*, they find that heart size is enlarged at P1 with increased CM size and CM proliferation. Deletion of *miR-128* from *Nkx2-5Cre* lineage during cardiogenesis did not affect heart function at P7, although CM size was decreased and CM cell cycle activity was increased at P7 and P14. Using a bioinformatics approach, authors search for predicted target genes underexpressed in adult heart relative to

neonatal heart and identify *Suz12* as a potential direct *miR-128* target. The direct regulation of *Suz12* mRNA by *miR-128* is shown in in vitro assays, and knockdown of *Suz12* in *Nkx-25Cre*; *miR-128* mutants rescues the proliferation phenotype. The authors then go on to test if *miR-128* may interfere with cardiac regeneration after injury, using the neonatal apical resection model.

In cardiomyocyte *miR-128* overexpression mice, neonatal hearts do not regenerate to the same extent as in control mice, and cardiomyocyte proliferation is decreased. Furthermore, it is shown that after myocardial infarction in adult mice, loss of *miR-128* enhances cardiac function and suppresses tissue fibrosis.

Reviewer comment 1: Previous work has shown a direct regulation of *Suz12* by *miR-128* (Peruzzi et al., *Neuro Oneal* 2013), it is suggested to cite this paper. Moreover, work from the Pu lab has demonstrated the relevance of the PRC2 complex for cardiomyocyte development (He et al., *Circ Res* 2012). Since this study also uses *nkx2-5Cre* to interfere with PRC2 complex function, it is recommended authors discuss this paper.

Author's Reply: As suggested, the paper (Peruzzi et al., *Neuro Oneal* 2013) has been cited on page 11 as follows:

‘Co-transfection of HEK293T cells with the *Suz12* 3’UTR plasmid (WT) and *miR-128* mimic resulted in a significant decrease in luciferase activity compared with cells co-transfected with the negative control or the mutated 3’UTR target sequence (Mut), indicating that *Suz12* is a direct target of *miR-128*, consistent with a previous report²².’

.’

Moreover, the paper by He et al., was cited and discussed on page 10 as follows:

‘Moreover, the analysis of genes downregulated in *miR-128*^{OE} showed statistically significant enrichment of genes downregulated after siRNA inhibition of components of polycomb repressive complex 2 (PRC2), *Suz12* in particular (**Supplementary Fig. 7C**). PRC2 is a major epigenetic modifier that affects multiple genes and is crucial for organogenesis. Perturbation of the epigenetic landscape during early cardiac development inhibits CM proliferation, and eventually leads to fatal cardiac malformations^{20, 21}.’

Reviewer comment 2: For neonatal hearts and isolated CM, the authors propose that *miR-128* suppresses *Suz12* leading to decreased p27, cyclinE and Cdk1 in neonatal hearts and isolated CM. However, to evaluate cell cycle activity in *miR-128* iKO heart after MI, the authors decide to use a different panel of genes compared to previous experiments in the manuscript. Can the authors show p27, CyclinE and Cdk2 levels in mutants versus controls? What happens to *Suz12* levels in this experiment?

Author's Reply: As suggested, Western blotting was performed for the expression levels of SUZ12, p27, CyclinE, and Cdk2 in the *miR-128* iKO hearts after MI. New data has been added in **Fig. 8B**. Analysis of iKO hearts at day 7 after TAM injection demonstrated that the expression of *miR-128* target (SUZ12) was significantly increased, accompanied by downregulation of cell cycle inhibitor (p27) and upregulation of cell cycle activators (CyclinE and CDK2).

Reviewer comment 3: What is the baseline expression level of *miR-128* in the heart and which cell types is it expressed in?

Author's Reply: As shown in **Fig.1E**, *miR-128* expression was significantly increased during cardiac development. CMs and cardiac fibroblasts (CFs) were isolated to analyze the expression of *miR-128* in different cell populations of the heart. As shown in **Fig. 1G**, *miR-128* expression in CMs was significantly higher than in CFs.

Reviewer comment 4: The authors claim that *miR-128* regulates neonatal cell cycle withdrawal. However, for their overexpression and loss of function models, *miR-128* levels are manipulated from embryonic stages onwards. Is *miR-128* also expressed during development, and can the authors show at which time point during development *miR-128* levels are changed in their overexpression and loss of function models? An alternative explanation to the observation that CM size is reduced and cell cycle activity is enhanced in the loss of function model is that embryonic CM do not mature in the absence of *miR-128* and thus retain an embryonic or fetal proliferative phenotype. Previous studies using *nkx2-5Cre* to ablate the PRC2 complex subunit *Ezh2* indicate requirements of this complex for CM development. As such the current study does not formally prove that *miR-128* in heart directly regulates neonatal cell cycle withdrawal, as this observation may be secondary to embryonic onset defects. Can the authors comment on this?

Author's Reply: The *miR-128* expression pattern indicated by qPCR during development in overexpression (*miR-128*^{OE}) and loss of function (*miR-128*^{-/-}) mouse models is now shown in **Supplementary Fig. 2E** and **Fig. 3B** respectively. By embryonic day 10.5 (E10.5), *miR-128*^{-/-} heart exhibited downregulation of *miR-128* expression. In addition, marked overexpression of *miR-128* in *miR-128*^{OE} hearts was also observed by E10.5.

Both *in vitro* and *in vivo* studies were performed to investigate whether *miR-128* is a negative cell cycle regulator. *In vitro*, we demonstrated that silencing of *miR-128* induces neonatal CM proliferation as indicated by significantly increased pH3⁺ CM, Aurora B⁺ CM, and EdU⁺ CM (**Supplementary Fig. 4**). *In vivo*, different transgenic mouse models were employed to further investigate the effect of *miR-128* deletion on CM proliferation. Importantly, cardiac specific, tamoxifen (TAM) inducible *miR-128* knockout mice were then generated by crossing α -MHC^{MerCreMer} mice with *miR-128*^{fl/fl} mice. TAM was administered at P21 to induce *miR-128* knockout at adult stage. As shown in **Figs. 7 and 8**, deletion of *miR-128* increased CM proliferation and promoted adult cardiac regeneration.

Reviewer comment 5: Authors report that *miR-128* mice display dilated cardiomyopathy consistent with focal replacement fibrosis, CM hypertrophy and severe heart failure in adults. Can data be provided to substantiate these claims?

Author's Reply: We have removed the comment regarding dilated cardiomyopathy in adult mice (which is apparent) and plan to address this in a follow up publication. New data was provided in **Supplementary Figs. 2F-H and Supplementary Fig. 3** to substantiate the claims of cardiac hypertrophy. As shown in **Supplementary Fig. 2F-H**, the *miR-128*^{OE} mutant mice at P28, displayed enlarged heart chamber, myocardial fibrosis, CM hypertrophy, and impaired left ventricular systolic heart function. Moreover, the KEGG pathway analysis based on our RNA-Seq data revealed that differentially expressed genes in *miR-128*^{OE} hearts were enriched in hypertrophic cardiomyopathy and dilated cardiomyopathy pathways (**Supplementary Fig. 3**). Further study by our lab is underway to explore the role of *miR-128* activation in dilated cardiomyopathy.

Reviewer comment 6: EdU incorporation is quantified by counting the number of EdU+ CM per section. Because the authors also report a change in cell size between conditions, these quantifications should be corrected for the total number of CM per section.

Author's Reply: As suggested, these quantifications were corrected and now calculated as percentage (%) of cells and the number of CM pool we analyzed was added in the figure legend accordingly.

Reviewer comment 7: Authors use a data mining approach to predict *miR-128* target genes that are under expressed in adult heart versus neonatal heart. Which dataset was used to establish differential expression?

Author's Reply: RNA-Seq dataset of control and *miR-128*^{OE} hearts was used to establish a differential expressed gene set. Accordingly, the following statement regarding the identification of *miR-128* target genes has been added on pages 9-10:

'RNA-Seq was performed on control (Ctrl) and *miR-128*^{OE} hearts (P7) to identify the putative target genes of *miR-128* responsible for cell cycle regulation. By comparing the downregulated mRNAs identified in *miR-128*^{OE} hearts relative to Ctrl hearts with all possible predicted candidate *miR-128* target genes¹⁹, we found 87 genes that contained the predicted binding site at the 3'UTR (**Supplementary Fig. 7A**). Gene Ontology (GO) PANTHER Analysis was then performed to identify the affected cellular biological processes. The leading biological category was 'cellular process' category, with nearly 28.7% of all associated genes (GO: 0009987, **Supplementary Fig. 7B**). A further subgroup analysis of the 'cellular process' indicated the potential for *miR-128* to affect multiple pathways that are related to regulation of the cell cycle, cell communication, and cellular component movement (**Supplementary Fig. 7B**). Moreover, the analysis of genes

downregulated in *miR-128*^{OE} showed statistically significant enrichment of genes downregulated after siRNA inhibition of components of polycomb repressive complex 2 (PRC2), *Suz12* in particular (**Supplementary Fig. 7C**). PRC2 is a major epigenetic modifier that affects multiple genes and is crucial for organogenesis. Perturbation of the epigenetic landscape during early cardiac development inhibits CM proliferation, and eventually leads to fatal cardiac malformations^{20, 21}. Significantly, *Suz12* was identified among the predicted downstream target genes of *miR-128*. In contrast to neonatal hearts, the protein levels of SUZ12 was lower in the adult heart (where the CM proliferation ability is quite limited) (**Figs. 4B-C**), paralleling the upregulation of *miR-128*. These data suggested a potential interaction between *miR-128* and *Suz12*, and was a key factor in generating our hypothesis that *miR-128* regulates CM proliferation in part through the PRC2-*Suz12* signaling pathway.'

Reviewer comment 8: Resolution of immune images is not great- for instance for figure 1c, Figure 2i, Figure 6b: it cannot be appreciated whether nuclei are CM nuclei or adjacent fibroblasts; for 6b: are we to believe that nearly all EdU+ cells are CM?

Author's Reply: As suggested, images in previous **Fig. 1C, 2I, and 6B** were replaced. **Fig. 7B** (previously **Fig. 6B**) shows triple immunostaining of EdU (red), cTnT (CM marker, green), and DAPI (nuclei, blue) to identify DNA synthesis in CM. Only triple positive cells were counted (at high magnification) as indicated with the white arrows.

Reviewer comment 9: The exact genotype and treatment of controls is often missing, making it impossible to evaluate if the right centers were used. Please specify the genotype, and treatment (for instance with/without tamoxifen) of control and mutant for each experiment, for instance in figure legends.

Author's Reply: As suggested, the specification of the genotype, treatment of the control group, and mutant for each experiment has been added in the **figure legends** accordingly.

Reviewer comment 10: The rescue experiment using *Suz12* siRNA in *miR-128* mutant hearts is not well described, please provide details.

Author's Reply: *In vivo* demonstration of the specificity of *miR-128-Suz12* signaling was achieved by conducting siRNA-mediated knockdown of *Suz12* in a *miR-128*^{-/-} heart (**Fig. 5D-H**).

Accordingly, the following statement was added on page 12:

'To further validate that the *Suz12*-pathway is a major functional mediator of *miR-128* effects, we injected *miR-128*^{-/-} mice (intraperitoneal (IP) injection) with si-*Suz12* or si-Ctrl at P1, P3, P5, and harvested the hearts at P7 (**Fig. 5D**).

Knockdown of *Suz12* *in vivo* significantly induced CM hypertrophy (Fig. 5E) and impaired CM proliferation by decreasing the number of EdU⁺ CMs (Fig. 5F). Moreover, there was a significant increase in the level of p27 (cell cycle inhibitor) and decrease of Cyclin E and CDK2 (cell cycle activator) expression in the si-*Suz12* group when compared with si-Ctrl treated hearts (Fig. 5G-H). These data indicate that *miR-128* deletion stimulates proliferation of CMs, in part through epigenetic modulation of cell-cycle related genes via targeting of *Suz12* (Fig. 5I).'

Reviewer comment 11: The authors should discuss the recent paper by Zeng et al. showing that *miR-128* inhibition during IR induced injury blocks cardiomyocyte apoptosis through the regulation of PPARG. What happens to apoptosis and PPARG in the studies presented in the current manuscript?

Author's Reply: As suggested, the recent publication by Zeng et al. has been cited.

Accordingly, the following statement was added on page 15:

'Although it was previously reported that *miR-128* regulates apoptosis by targeting peroxisome proliferator-activated receptor gamma (*PPAR* γ)²³, we found no significant differences in either *PPAR* γ expression or apoptosis in iKO hearts when compared to Ctrl hearts at day 7 after TAM injection (Supplementary Figs. 10C and D).'

REVIEWERS' COMMENTS:

Reviewer #2 (Remarks to the Author):

The authors addressed most concerns, but there is one issue remaining: in figure 3L the authors show Ki67/ EdU double staining. However, the Ki67 signal is very diffuse and looks more like a cardiac protein staining. Please clarify whether indeed Ki67 is shown and provide novel data for a Ki67 staining with adequate distribution.

Reviewer #3 (Remarks to the Author):

The authors attempted most issues raised and in doing so were able to substantially improve the quality of the manuscript. There are no further comments

To Reviewer # 2: We appreciate the reviewer's comments, which were very helpful in improving the manuscript.

Reviewer comment: The authors addressed most concerns, but there is one issue remaining: in figure 3L the authors show Ki67/ EdU double staining. However, the Ki67 signal is very diffuse and looks more like a cardiac protein staining. Please clarify whether indeed Ki67 is shown and provide novel data for a Ki67 staining with adequate distribution.

Author's Reply: We appreciate the reviewer's comment. Indeed, the immunostaining of cTnT (Green color) in Fig. 3M (previously Fig. 3L) had been mislabeled as Ki67. We apologize for this careless mistake. As suggested, we have corrected this oversight and replaced 'Ki67' with 'cTnT' in this figure.

My colleagues and I hope that this concern of the reviewer has been fully and satisfactorily addressed. Thank you again for your consideration.